# Risk Assessment and the Effects of Refuge Availability on the Defensive Behaviors of the Southern Unstriped Scorpion (*Vaejovis carolinianus*)

**DOI:** 10.3390/toxins12090534

**Published:** 2020-08-20

**Authors:** David R. Nelsen, Emily M. David, Chad N. Harty, Joseph B. Hector, Aaron G. Corbit

**Affiliations:** Department of Biology and Allied Health, Southern Adventist University, 4881 Taylor Cir, Collegedale, TN 37315, USA; emilydavid@southern.edu (E.M.D.); chadharty@southern.edu (C.N.H.); josephh@southern.edu (J.B.H.)

**Keywords:** venom optimization, venom metering, hide, retreat, sex differences in behavior, THREAT assessment

## Abstract

Selection should favor individuals that acquire, process, and act on relevant environmental signals to avoid predation. Studies have found that scorpions control their use of venom: both when it is released and the total volume expelled. However, this research has not included how a scorpion’s awareness of environmental features influences these decisions. The current study tested 18 *Vaejovis carolinianus* scorpions (nine females and nine males) by placing them in circular arenas supplied with varying numbers (zero, two, or four) of square refuges and by tracking their movements overnight. The following morning, defensive behaviors were elicited by prodding scorpions on the chelae, prosoma, and metasoma once per second over 90 s. We recorded stings, venom use, chelae pinches, and flee duration. We found strong evidence that, across all behaviors measured, *V. carolinianus* perceived prods to the prosoma as more threatening than prods to the other locations. We found that stinging was a common behavior and became more dominant as the threat persisted. Though tenuous, we found evidence that scorpions’ defensive behaviors changed based on the number of refuges and that these differences may be sex specific. Our findings suggest that *V. carolinianus* can assess risk and features of the local environment and, therefore, alter their defensive strategies accordingly.

## 1. Introduction

The ability to acquire, process, and respond to internal (e.g., satiety, hormonal balance, venom availability, etc.) and external (density of resources and presence of potential predator, etc.) information is essential for survival and forms a route through which natural selection can drive evolution [1,2]. The more efficiently and accurately an organism can acquire, process, and appropriately respond to information, then the more likely the organism will be to survive and reproduce. Thus, natural selection should favor organisms that can integrate various forms of information and respond to their interpretations of the information in useful ways [3,4]. One important way that organisms do this is in the assessment of risk. To summarize Blanchard et al., risk assessment occurs when an organism calculates the probability of incurring harm when weighing behavioral options [5]. One well documented example within venomous organisms is venom metering [6], also known as venom optimization [7,8]. The use of venom and the decision related to how much venom to use are associated with various risks. The use of venom, either for defense or predation, requires direct contact with another organism, which increases the risk of injury due to retaliation. The use of too little venom may result in the loss of a prey item or may serve as an inadequate deterrent. The use of too much venom may negatively impact the energy budget, diverting resources that could be used for reproduction into venom regeneration and depleted reserves of venom may compromise future prey acquisition or self-defense. 

There is a sizable body of literature on venom metering, with several articles reviewing this topic in different groups of organisms (see [9] for venom use in spiders, [10] for venom use in scorpions; [6] for venom metering in snakes, [8] for a general review). The combined results of these studies suggest that the use of venom is a tightly controlled process, both in the choice to use or not use venom and in the precise controlling of the venom’s quantity expressed when used. Venom use has been observed to vary with level of satiety [11], venom availability [12,13], prey size [14,15,16,17,18], prey struggle intensity [7,18,19], prey type [19,20,21], type of threat [22], persistence of threat [23], and degree of threat [6,24,25,26].

Venom metering has been well documented within scorpions. Starting in the 1980s, it was observed that scorpions modulate their use of stinging for prey capture. Two studies [27,28] reported that stinging occurred more often in younger scorpions compared to older. Since these initial studies, scorpions have also been observed to modulate sting use in conjunction with prey size [29,30], prey state (i.e., alive/dead) [29], prey struggle intensity [30], prey type [31], and the mass of the stinging scorpions (in other words, larger scorpions stinging less frequently) [32]. Males and females have been observed to use stinging behavior differently, although when size was controlled, both males and females inject similar quantities of venom [33]. Additionally, scorpions release venom more often when the perceived threat is higher [25,34], scorpions prefer to hunt smaller prey when venom availability is limited [35], and scorpions deliver a larger quantity of venom when under greater duress [25,34,36,37]. Interestingly, one study reported that, in some circumstances, scorpions sting less and use less venom when the threat is greater; however, in this case, the threat persisted over longer periods of time [38]. 

Most studies that investigate venom use do so in the context of venom metering; however, venom use is only one of many behavioral options that venomous organisms may use in the context of predation or defense. When acting in defense, a venomous organism may employ other behaviors aside from venom use. Less research has examined this larger context, but those that do generally show that venomous animals will choose defensive behaviors to incur less risk before they resort to venom use. Several studies on snakes have found that, during simulated threatening events, snakes are more likely to use a variety of defensive behaviors (e.g., fleeing, threat display, remaining motionless, deploying musk) rather than strike [39,40,41,42]. In spiders, Nelsen et al. showed that the western black widow spider (*Latrodectus hesperus*) generally remains motionless initially, then flees, then attempts other behaviors like thanatosis and deploys defensive silk, before it escalates to biting [26]. Scorpions also seem to follow this trend. Carlson and Rowe reported that, for *Centuroides vittatus*, it was much easier to elicit fleeing behavior than stinging behavior when prodding them on the metasoma or the dorsum of the prosoma [43]. van der Meijden et al. surveyed the defensive behavior of 26 species across at least six families and found that, despite differences between taxa, the utilization of stinging was very common [44]. However, this study did not stratify according to threat and, therefore, did not address the question of risk assessment. Recently, Lira et al. utilized the technique reported by van der Meijden et al. to elicit defensive behavior in *Tityus pusillus* and did stratify by threat level [44,45]. This study found that stinging was much more likely when the scorpion was grasped by the prosoma, rather than by the chelae.

When considering risk assessment in scorpions, both studies were limited to venom metering. Studies that include other defensive behaviors indicate that scorpions can use numerous inputs, both internal and external, to assess risk and modulate behavior. However, several gaps in available knowledge exist. Firstly, studies that examine this topic are highly skewed towards the family Buthidae and very few other families have been tested. Of 13 studies that looked at venom metering in scorpions, eight examined species within Buthidae and the only two relevant studies that looked at other defensive behaviors were also focused on Buthids [43,45]. Other scorpion families are thus understudied. Secondly, the kinds of information inputs that scorpions use to assess risk and modulate behavior is little understood. Previous studies focus on internal factors (e.g., hunger) or immediate external stimuli (e.g., prey struggle intensity, persistence of threat). However, there are no studies that look at whether scorpions use other external information outside of the immediate stimulus (e.g., general environmental awareness) to modulate their behavior. The presence of refuges has been shown to alter defensive behavior in fish [46], and it is logical to assume that this would also extend to other animals. Terrestrial animals have a variety of senses (e.g., vision, olfaction) that enable them to understand the features of their present environment, which allows them to assess risk and to modulate behavior in other contexts. For example, the western Diamondback Rattlesnake (*Crotalus atrox*) can use its heat-sensitive labial pits to find warmer or cooler areas in its immediate environment, which can modulate its thermoregulatory behavior [47]. Hamsters, birds, honeybees, and desert ants are also known to utilize landmarks as inputs to modulate their navigational behavior [48]. 

Though studies are not conclusive, scorpions may also be able to modulate their behavior based on awareness of local environmental features. The evidence for this is also in the context of navigation. Gaffin and Brayfield have suggested that *Paruroctonus utahensis* may use scene familiarity (as defined by [49]) to navigate back to their burrows after a bout of foraging [50]. Given that scorpions may be able to use their awareness of the features in their immediate environment in the context of navigation, we hypothesized that they may also use this kind of information to modulate defensive behaviors. Scorpions may have an awareness of places in their environment where they can shelter from predators, either through learning or from immediate sensory cues. If refuges are available, then a scorpion may be more likely to flee to a refuge rather than utilize venom or other behaviors when under threat. 

In this study we investigated the defensive behaviors of the southern unstriped scorpion, *Vaejovis carolinianus* (Beauvois 1805, Vaejovidae). We examined how they modulate their defensive behaviors based on which of three body segments were prodded; how this species uses their metasoma for defense in relation to other behavioral options; and if the presence and number of refuges influences their defensive behaviors. Thus, this study seeks to expand our knowledge in three areas. First, this study investigates risk assessment in a previously unstudied family of scorpions. Second, our research expands the understanding of venom use and stinging in the context of other defensive options. Lastly, this study investigates if awareness of features in the immediate environment modulates defensive behaviors.

## 2. Results

We tested 18 scorpions (N = nine females and N = nine males), by placing them in circular arenas supplied with varying numbers (zero, two, or four) of square refuges and by using video to track their movements during an overnight acclimation period. Except for one male, that was lost after the first test, we tested each scorpion under each refuge condition with the order of treatments randomized. We have provided a list of key terms and definitions in Table 1. The scorpions showed substantial variation in their movement patterns during the acclimation period (Appendix A). Mean movement across all trails was 1.05 cm/s (SD = 0.43, N = 50). We did not detect a relationship between mean movement and the number of refuges (Linear Mixed Model (LMM), *F* = 1.26, *df* = 2, *p* = 0.308) or sex (LMM, *F* = 0.42, *df* = 1, *p* = 0.528). 

We measured the combined length of the prosoma and mesosoma for each scorpion. We found significant size dimorphism between sexes (*t* = 7.90, *df* = 16, *p* < 0.001; Figure 1). 

After the acclimation period, we assessed each scorpion’s defensive behavior by prodding it 30 times: 10 times, once per second, on the chelae; 10 times on the prosoma; and, finally, 10 times on the distal metasoma (excluding the telson). We subjected each scorpion to three cycles. The trial ended either when all three cycles were completed or when the scorpion retreated under a refuge, whichever came first. For 43 of 52 trials, the scorpion was observed for the full duration of all three cycles (90 s). In the remaining nine trials, the observation time was reduced because the scorpion entered a refuge. For the reduced trials, the total observation time varied between 11.97 and 87.43 s. 

### 2.1. General Use of Defensive Behaviors

Stinging was a relatively common behavior. Median scorpion stings per trial was 37 (range: 0–79), while median pinches per trial was only 0.5 (range: 0–9). The scorpions stung at least once in 50 of 52 trials (96.1%) compared to only pinching in 26 of 52 trials (50.0%). During the trials when scorpions stung, we observed venom use in 26 trials (52.0%). Fleeing was also a common behavior and the scorpions exhibited this behavior in all trials. The median duration of fleeing per trial was 18.98 s (range: 1.50–51.90).

Fleeing was often the first observed behavior. The median latency to flee was 3.97 s (range: 0.03–23.13) compared to 13.50 s (range: 1.00–35.93) for stinging and 27.12 s (range: 2.00–85.13) for pinching. Overall, median time to the first recorded behavior of any kind was 3.52 s (range: 0.03–22.2). In the trials that included the no-refuge condition, fleeing was the first behavior for 14 of the 18 scorpions (77.8%) and the other four scorpions (22.2%) stung first (*χ*^2^ = 5.56, *df* = 1, *p* = 0.018). For the four-refuge condition trials, the results were similar. In this case, fleeing was the first behavior of 13 of 17 scorpions (76.5%); however, three other scorpions stung first (17.6%), and one (5.9%) pinched first (*χ*^2^ = 14.59, *df* = 2, *p* = 0.001). The two-refuge condition result was unique in the series of trials. In the two-refuge condition, the first recorded behaviors of three of the scorpions (two males and one female) were simultaneous attempts to flee and to sting. When these three scorpions were removed from the analysis, we found that 11 of the 14 (78.6%) remaining scorpions fled first and the three remaining scorpions (21.4%) stung first (*χ*^2^ = 14.57, *df* = 1, *p* = 0.033).

### 2.2. Results of Mixed Model Analysis

#### 2.2.1. Cycle

Table 2 shows the results of the LMM and generalized linear mixed model (GLMM) analysis of cycle. A full description of every LMM and GLMM including all coefficients and associated confidence intervals, can be found in Appendix A. These results demonstrated that the cycle was significant in the sting frequency and flee duration models (*p* < 0.001 for both), but was not for pinch frequency or venom use. Post hoc tests revealed that scorpions stung more in the second cycle than the first or third (*p* < 0.001); however, the number of stings for the first and third cycle were similar (Figure 2A). Post hoc tests also found that flee duration was shorter for the third cycle than for the first or second cycles (*p* < 0.001 and *p* = 0.016, respectively), but flee duration was similar for the first and second cycles. However, marginal means (Figure 2B) suggested a trend that flee duration decreased as cycle increased. 

#### 2.2.2. Prod Location

Prod location was significant for all models that it was included in (*p* ≤ 0.011 for all models; Table 3). Post hoc tests showed that sting frequency differed between each prod location (*p* < 0.001 for all) with prosoma > metasoma > chelae (Figure 3A). The pattern was different for pinch frequency. Although prosoma and chelae were similar, the metasoma differed from both prosoma and chelae (*p* = 0.008 and *p* < 0.001, respectively) with prosoma = chelae > metasoma (Figure 3B). Flee duration was similar when stimulation was to the metasoma or prosoma; however, both were different from the chelae (*p* < 0.001 for both) with prosoma = metasoma > chelae (Figure 3C). Although, chelae and metasoma stimulation yielded similar numbers of venom use events, prosoma differed from both metasoma and chelae (*p* = 0.013 and *p* = 0.023, respectively) with prosoma > metasoma = chelae (Figure 3D).

#### 2.2.3. Number of Refuges and Sex

In the trails, the number of refuges was significant only for the latency to pinch (*p* = 0.007). In this case, the two-refuge condition was different from both the no-refuge and four-refuge condition (*p* = 0.048 and *p* = 0.004, respectively). In the no- and four-refuge condition, scorpions took longer to pinch than those in the two-refuge condition (Figure 4A). However, the number of refuges was nearly significant (*p* = 0.060) for the model assessing pinch frequency. Here, post hoc tests showed that, while the no- and two-refuge condition were similar, both had a greater pinch frequency than the four-refuge condition (*p* = 0.018 and *p* = 0.033; Figure 4B). 

The main effect of sex was not significant in any model; however, it was close to the significance threshold (*p* = 0.090) in the model for venom use. The model suggested that males were observed to use their venom only 22% (95% CI, 4–127) as often as females.

We found significant interactions between the number of refuges and sex in both the sting frequency and venom use models (*p* = 0.003 and *p* = 0.048, respectively, Table 3). Post hoc analysis suggested that females stung more in the four-refuge and two-refuge condition than when there were no refuges (*p* = 0.002 and *p* = 0.026, respectively). However, statistical differences were not detected within male refuge conditions or between males and females in any refuge condition (Figure 5A). For venom use, post hoc analysis suggested that females used venom more in the four-refuge condition than males in the four-refuge condition or males in the no-refuge condition (*p* = 0.038 for both, Figure 5B). We did notice that the refuge number by sex interaction was close to significance for the latency to pinch model (*p* = 0.073). Here, the post hoc tests suggested that males in the four-refuge condition took longer to pinch than males in the no- and two-refuge conditions (*p* = 0.036 and *p* = 0.008, respectively; Figure 5C). 

### 2.3. Covariate Controls

While scorpion length and observation time were not significant in any model, mean movement during the overnight acclimation period was significant in the models assessing sting frequency (*p* < 0.001), venom use (*p* = 0.009), and latency to sting (*p* = 0.026). The models indicated a 27% increase in stinging (95% CI, 14–42%), a 77% increase in venom use (95% CI, 15–172%), and a 2.10 s reduction in sting latency (95% CI, 0.33–3.87 s) for every unit increase in mean movement. 

## 3. Discussion

In this study, we sought to test whether *V. carolinianus* modulates defensive behavior according to which of the three body locations were prodded; how the scorpions use their metasoma for defense in relation to other possible defensive behaviors; and if the presence and number of refuges within their environment also influenced defensive behavior. We found strong evidence that these scorpions perceived prods directed at different body segments as more threatening than prods to other body parts. In fact, prods to the prosoma corresponded with increased use of all defensive behaviors. We also found that although stinging is a common defensive behavior in this species, its use typically occurred after the simulated threat had persisted, and after other strategies, such as no reaction and flee, had been used first. Lastly, we found some indications that scorpions modify their behavior based on the presence of refuges, although it is unclear precisely how the presence of refuges affects defensive behaviors.

### 3.1. Risk Assessment Based on Prod Location

It appears that a touch to the prosoma is perceived as more threating than either the chela or the metasoma. Generally, we observed more of all defensive behaviors (sting, venom use, pinch, and flee) and the scorpions were quicker to react defensively when they were touched on the prosoma. Our results are similar to those reported by Lira et al., who also found that the scorpion, *T. pusillus*, used different behaviors depending on which body part was restrained [45]. *Tityus pusillus* responded more with fleeing or thanatosis when restrained by the telson and stinging, tail wagging, standing still or fleeing when restrained by the chela or prosoma. Lira et al. also reported that scorpions used stinging significantly more often when pinched by the prosoma than when pinched by the chelae. These results, although similar to ours, also present some differences. Like Lira et al., we found that stinging occurred significantly more often when the prosoma was touched; however, we observed more stinging when the metasoma was touched than when the chelae was touched. This difference may be a result of how the defensive behaviors were elicited. Lira et al. conducted two distinct trials: the first elicited defensive behaviors by grasping the telson with forceps and then dropping the scorpion from a fixed height. The second trial elicited defensive behaviors by grasping the chelae (one at a time in a random order), then by grasping the prosoma. We elicited defensive behaviors by prodding the scorpions in all three locations during a single observation and thus our design allows for a more direct comparison of risk assessment based on simulated predatory attacks to specific body regions than does Lira et al. In addition, it is important to note that Lira et al. did not statistically analyze sting use between the first and second trials, but merely commented on the most commonly observed behaviors in each.

Interestingly, Lira et al. reported that they observed more attempts to flee after scorpions were released from a grasp to the chelae compared to the prosoma. We, however, observed significantly more time spent fleeing as a result of prods to the prosoma and metasoma compared to the chelae. We found no difference in the time the scorpion spent fleeing between prods to the prosoma and metasoma. Again, this is likely due to differences in how defensive behaviors were elicited. It is possible that when a scorpion that is held by a chela, it may already begin to pull away from the grasp before it is released and, thus, it is more likely to continue to move when the grasp is released. However, holding a scorpion by the prosoma was more likely to result in stinging, which is more easily and accurately accomplished while the scorpion is stationary than during fast movements. This may lead scorpions to continue to sting or to assume a sting-ready posture when released, rather than for scorpions to initiate a new behavior like fleeing. The persistent nature of our method for eliciting defensive behaviors, however, allowed the scorpions to switch between several different defensive behaviors during the 10 s that they were repeatedly prodded on a specific body part. 

If we aggregate the Lira et al. study [45] with our results and the other literature, we can conclude that stinging and venom use are behaviors scorpions typically reserve for the most dire situations (see introduction). This evidence suggests that scorpions treat attacks to their prosoma as a greater threat than attacks to the chelae or metasoma. We have observed scorpions both in the field and as a result of experimental manipulation survive damage to, or the loss of, a chela or telson. In fact, some species of scorpions can autotomize large portions of their metasoma to escape predators and are reported not only to be able to survive without their metasoma for several months, but also to mate successfully [51]. To many arachnids, damage to the prosoma may be more life threatening, because the prosoma has a very high internal pressure that is used to facilitate locomotion; however, if the prosoma becomes perforated, then arachnids become susceptible to rapid and extensive bleeding [52,53]. The prosoma also contains, or is close to, many vital organs [54,55]. 

We found that *V. carolinianus* was more likely to sting than to pinch with their chela during defensive encounters. Pinching occurred more frequently when the scorpions were prodded on the chelae or prosoma than on the metasoma. This may indicate that chelae use is a behavior of convenience (proximity) and is used more to aid attempts to sting than as an independent and isolated defensive behavior. Thus, it appears that this species relies more heavily on stinging to deter a threat than on their chelae. Similar to our observations, Van der Meijden et al. 2013 found a correlation between the morphology of chelae, metasoma, and their use for defense [44]. Species that had strong chelae tended to use their chelae more for defense. Most of the species observed used both their chelae and telson for defense: those that used only their metasoma were also the species with the most medically relevant venom. The disproportionate use of stinging compared to grasping in *V. carolinianus* is likely due to the gracile nature of their chelae more than the potency of their venom. Although the venom of *V. carolinianus* has not been well studied, it is unlikely to be medically significant in humans due to the scorpion’s small size (and thus small venom reserves), in addition to the fact that the severity of envenomation has only been reported to range from local swelling to small, localized necrotic lesions, similar to that of the sting from a honey bee [56,57]. 

Though not statistically significant, our data did indicate that females released venom more often than males. Differences in defensive behavior between the sexes have also been reported by other studies [32,33,43,45], as has sex-specific differences in the venom composition of several scorpion species ([33] and references therein). Of the studies that reported behavioral differences most found that females stung more frequently [33], were quicker to sting (i.e., shorter latency) [32,33], and stung at a faster rate [32,43]. However, all these studies were conducted in the buthid scorpion *C. vittatus*, a sexually dimorphic species, and only one controlled for size [32]. However, this paper [32] found that scorpion size-related morphological differences (e.g., body mass, body length, metasoma length, metasoma mass, etc.) did not significantly influence stinging behavior. On the other hand, Miller et al. found that although females deliver larger absolute quantities of venom, when mass was controlled for this, statistical difference disappeared [33]. Thus, previously published literature does not clarify whether size, sex, or both contribute to differences in defensive behavior and venom expenditure. The species we examined is also sexually dimorphic for overall body size: adult females are larger than males. While we did control for size on this study, we only detected a possible effect of sex on venom use and all other main effects were insignificant. This may reflect an observational bias due to the difficulty of seeing small droplets of venom. Larger individuals have larger venom glands and thus may also release proportionally larger droplets of venom than smaller individuals, which may have biased our observations against males. Even when our results are considered alongside other studies, the interplay of sex and size on defensive behavior and venom use in scorpions has yet to be disentangled. 

Despite the lack of clarity about the significance of a scorpion’s sex on stinging behavior, the mean overnight movement had a significant influence on stinging behaviors. We found that if individuals that were more active, they also stung significantly more often, were more likely to use venom compared to individuals that were less active, and tended to sting more quickly. In other words, the mean overnight movement may be a measure of an individual’s health. It would be predicted that healthier individuals would have more energy reserves and, therefore, would be more likely to continuously move about in their habitat and vigorously defend themselves using venom and associated behaviors. Scorpions in poorer condition may be less able to invest metabolic resources in venom production, which may lead them to have less venom and, therefore, to be less likely to use behaviors associated with venom. This interpretation is consistent with the venom metering hypothesis that bases its predictions on the ecological and metabolic cost of venom use and production [6]. However, few studies have calculated the metabolic costs of venom production. McCue found an 11% increase in the resting metabolic rate of three species of North American vipers (*C. atrox*, *Crotalus horridus*, and *Agkistrodon controtrix*) as a result of venom regeneration [58]. Similarly, two studies by Nisani et al. found a 39% and 21% increase in the resting metabolic rate of *Parabuthus transvaalicus* due to venom regeneration [59,60]. However, two additional studies reported no discernable increase in the metabolic rate of either the common death adder (*Acanthophis antarcticus*) [61] or the prairie rattlesnakes (*Crotalus viridis viridis*) [62]. Therefore, even though the only studies to investigate the metabolic cost of venom in scorpions both reported that venom regeneration was linked with a significant increase in the resting metabolic rate, the true metabolic cost of venom remains unclear and demands further investigation. 

Alternatively, the correlation of these defensive behaviors with activity level may be part of a behavioral syndrome, which is defined as a suite of correlated behaviors that persist across physiological and behavioral contexts [63]. In other words, the correlation of these behavioral traits may be a function of each scorpion’s personality rather than their physiological condition. The influence of personality and the existence of behavioral syndromes on behaviors associated with venom use has, to our knowledge, never been studied and is an obvious area for future work. 

### 3.2. Venom Use and Associated Behaviors in the Context of Other Defensive Behaviors

Our analysis of the latency to first behavior suggests a clear chronological pattern of defensive behaviors. We found that the most common initial response to our simulated threat was for a scorpion to not react. In fact, it took most individuals around 3.5 s, or three prods, before the scorpions responded in any way to the simulated threat. When the scorpions did finally perform a behavior, they were significantly more likely to flee. In general, our data point to a general pattern where, first, the scorpions did not react, then they attempted to flee, and, finally, they made use of behaviors associated with venom. This general pattern is in line with several previous studies [26,33,39,43,64]. As described above, most research that has investigated the use of venom-associated behaviors for defense do not typically also study how these behaviors are used in the broader context of other possible defensive behaviors. However, there is a limited body of literature across several taxa that suggests that behaviors associated with venom use are generally not the first behaviors used in response to a simulated threat; however, the use of venom may be employed quickly if the perceived threat is high. Glaudas et al. found that roughly 76% of individuals from a population of pigmy rattlesnakes (*Sistrurus miliarius*) did not react when they were approached and touched on the snout and remained in the position they were found in [39]. Of those individuals that did react, the most common response was to flee and not strike. Similarly, Herbert and Hayes found that it took between 1–4 min of persistent testing before individual southern pacific rattlesnakes (*Crotalus helleri*) struck at a target under their simulated threat [64]. Nelsen et al. found that, like the scorpions used in this study, the western black widow spider (*L. hesperus*) initially responded to simulated threats, like those used in this study, either by not reacting, retracting the leg that was touched, or briefly fleeing from the threat [26]. However, as the threat persisted, these spiders would begin to use other defensive behaviors such as thanatosis, longer bouts of fleeing, defensive silk use (silk flicking), and biting; with the later behaviors typically appearing more as the threat persisted. This study also made use of several threat levels and found that prodding the spider resulted in much less biting than when the spiders were grasped, which suggested that the spiders found grasps to be much more threatening. Likewise, Carlson and Rowe noted that they had difficulty getting the scorpion, *C. vitattus*, to sting, but that getting them to run in response to their stimulus was comparatively easy [43]. Miller et al. noted that very few male *C. vittatus* scorpions stung at all during the first of five simulated attacks [33]; however, this trend was not observed in females. 

One the other hand, Nisani and Hayes appeared to be able to elicit venom spraying easily in the scorpion *P. transvaalicus*, which occurred an average of 0.33 s following stimulation [36]. Rasko et al. studied the giant desert hairy scorpion, *Hadrurus arizonensis*, and observed that a high proportion of individuals use venom and associated behaviors immediately. Moreover, Rasko et al. were surprised to find that stinging, instances of venom use, and the volume of venom delivered all decreased with persistence of threat [38]. However, both the Nisani and Hayes study and the Rasko et al. study forcefully restrained their study animals. Nisani and Hayes grasped the scorpions by the metasoma, while Rakso, et al. pinned the scorpions by the prosoma for 10 s. Based on Nelsen et al., this likely represented a highly stressful threat that was more likely to produce stinging and venom use than a less stressful threat such as a prod. This may have caused these scorpions to employ their venom immediately, rather than utilizing their other defensive behaviors. The decrease in stinging and venom use described by Rasko et al. may have been due to fatigue or acclimation to the simulated threat. 

If these studies are assessed together, then it appears that venom use and associated behaviors are not immediately employed for defense, especially if the threat is perceived as more mild. The use of venom appears to be reserved for greater threats, as determined by the force or anatomical location of the threatening stimulus or the persistence of the threatening stimulus over time. This is consistent with the conclusion that an organisms’ assessment of risk influences their behavior, and thus, adds further support to the venom metering hypothesis.

### 3.3. Changes in Defensive Behavior across Cycle

Across the three cycles of our study, the frequency of stinging in the second cycle was significantly greater than either the first or third cycles and we found a general trend of decreasing flee duration. We observed no significant differences between cycles for pinch frequency or venom use observations, likely because these behaviors were both relatively rare. These results are consistent with the venom metering hypothesis. As stated above, scorpion behaviors associated with venom delivery are often reserved for situations deemed to be more threatening; therefore, we would expect an increase in their use as the threat persists and other behaviors, such as fleeing, would be utilized much sooner. The decreases in the scorpions’ flee duration and stinging subsequent to peak use could represent either habituation or fatigue, but further testing is necessary to disentangle these possible causes.

### 3.4. Presence and Number of Refuges

In addition to investigating the presence of risk assessment and its effect on defensive behaviors, one of our primary research goals was to investigate if *V. carolinianus* used environmental features when making decisions about how to behave. In this case, we were interested in discovering if the presence and number of available refuges influenced defensive behaviors. Our results suggest that this species may glean information about its surroundings and that this information impacts its defensive strategies. We found that scorpions took less time to pinch in the two-refuge condition than for the no- or four-refuge condition. Although it is not significant, our data also suggested that our scorpions pinched less in the four-refuge condition than in the no- or two-refuge condition. Our data also suggest that males and females may respond to information about the surrounding environment differently. Females generally stung more often as the number of available refuges increased, while males showed no discernable trend. Likewise, females were more likely to use venom when more refuges were available, unlike males under the same condition. Though not statistically significant, our data also may suggest that males took longer to pinch when more refuges were present, but females showed no discernable trend. 

We find these results puzzling. We are not aware of any biological theory that would explain why scorpions would take less time to sting when two refuges were present compared to no- or four-refuges. Animal behavior can be highly variable, and our sample size was relatively small (N = 17 complete observations with one additional single observation); therefore, future studies will benefit from an increased sample size to help detect true signals reliably. 

Although the overall reduction in pinch frequency and the increase in latency to pinch for males in the four-refuge condition matches our initial prediction, the increase in stinging and venom use among females did not. Initially, we assumed that scorpions would be more likely to flee toward a refuge when they were readily available and be more likely to sting when there were fewer places to hide. Our results present two major questions: why would more refuges lead to more stings and venom use? Why were there sex-specific responses? Concerning the increased use of sting and venom, previous work has shown that scorpion venom is divided into two main types: the salt-rich, pain inducing pre-venom and the proteinaceous main venom [65]. This pre-venom may be the primary venom used in defensive behavior and may cause a potential predator to recoil in pain allowing time for the scorpion to escape. This may be true even when scorpions encounter predators that are resistant to their venom, such as grasshopper mice (*Onychomys* spp.). In studies that have observed interactions between these species and bark scorpions (*Centruroides* spp.), the initial stings caused the mouse to release the scorpion and vigorously groom itself [66] which, in more natural conditions, may have allowed the scorpions enough time to escape, especially if the scorpion was near a refuge. This may suggest a hypothesis that if a scorpion is aware that a refuge is nearby, they may increase their use of venom and associated behaviors in order to aid their attempts to flee toward a refuge. 

It is not immediately clear to us why scorpions may differ in their response to environmental conditions between sexes; however, our results were not without precedent. As described above, several studies have found that males and females prefer different defensive behaviors [32,33,43,45]. In addition, male *C. vittatus* climb higher in trees than females [67] and males in the genus *Ananteris* are more likely to autotomize their metasoma than females [52]. Of note, *V. carolinianus* has been reported to prefer sex-specific microhabitats: females are found under and among rocks, as well as at the base of dead standing trees and males in leaf litter and under the bark of dead pine trees [68]. If examined together, these studies may suggest that male and females prioritize different environmental information, but the nature of these differences and the implication of size dimorphism remains a mystery in need of further investigation. 

Because our study represents one of the first times anyone has investigated if and how features of the immediate environment influence the venom metering of scorpions, more research is needed. The possible interpretations provided are not exhaustive and serve only to provide inspiration for future research. The more plausible interpretations will only become clear as more research tests hypotheses and rules out erroneous assumptions.

## 4. Conclusions

Our research presents evidence that a species in the family Vaejovidae are able to modulate their defensive behavior based on perceived risk and, as such, this study provides a new line of evidence that supports the concepts of risk assessment and venom metering from an understudied branch of the scorpion phylogenetic tree. This research also provides evidence that scorpions may have a general awareness of their surroundings that can influence their defensive behaviors. However, these initial findings need to be verified by future research. If the behavior of these organisms is not only influenced by an immediate stimulus, but also by a general awareness of their immediate environment, then it would suggest they possess a higher degree of cognitive complexity than previously known.

## 5. Materials and Methods

### 5.1. Collection

We collected male and female scorpions from September through early October of 2018 on the campus of Southern Adventist University in Collegedale TN USA (35.0482° N, 85.0520° W). The campus includes a broadleaf temperate deciduous forests and scorpions were collected at the border between the forest and the central campus. In total, we collected 23 adult scorpions (10 male and 13 female). Three of the female scorpions were used in proof of concept trials and were not included in the study. Two scorpions died during the study and one male was lost after testing it the first time, which resulted in our ability to have 17 observations for the two- and four-refuge conditions (8 male and 9 female) and 18 observations for the no-refuge condition. 

### 5.2. Care

Scorpions were housed in 500 mL plastic deli containers (TRiPAK TD40016) half-filled with All Purpose Garden Soil (Miracle-Gro^®^ item# 70551430). We maintained the scorpions at a constant temperature (23.9 °C) and a constant light/dark cycle consistent with the natural light/dark cycle for December in the Southeastern USA (Sunrise 7:30 a.m. and sunset 5:30 p.m. EST). We fed the scorpions a three-week-old cricket (*Acheta domesticus*) once per week and removed any uneaten food two days later to prevent bacterial or mold growth. The scorpions had access to water ad libitum. All scorpions were returned to the conditions described above once the study was concluded. 

### 5.3. Testing Arenas for V. carolinianus Risk Assesment Trails: (A) no, (B) Two, and (C) Four Refuge Condition

For the arenas, we used four 63.5-cm diameter circular plastic plant saucers (GroPro Part# 724946) that were filled with a layer of moistened All Purpose Garden Soil, just thick enough to cover the entire base of the arena. To vary the mock habitats, the arenas either had no refuges, two refuges, or four refuges available for scorpions to explore and hide under (see Figure 6). The refuges were square pieces of cardboard (104 mm × 104 mm) with the corners slightly flexed down so that there was a sufficient gap between the refuge and substrate for the scorpions to walk under. We commonly find *V. carolinianus* under rocks in their natural habitat and under boxes, blankets, etc. in buildings. Our refuges approximated the conditions this species is commonly found under, both natural and man-made, and permitted us to standardize the refuge size and use new refuges for each trial. 

### 5.4. Experimental Randomization/Standardization

We tested each scorpion three times, once for each refuge condition (no-, two-, and four-refuges) and randomized the order of treatment. We standardized the scorpions’ feeding schedule so that each scorpion was fed one week prior to testing; therefore, each scorpion was tested approximately once per week over a three-week period, starting 26 November through 14 December 2018. Because we only had four total testing arenas, we removed used substrate, cleaned and dried the arena with 70% ethanol, and then added fresh substrate and new refuges between each trial in order to remove any chemical or physical information that may have been left by other scorpions.

### 5.5. Exploration of Arenas

Scorpions were acclimated to the arenas overnight before they were subjected to a simulated predatory encounter (see below). We placed the scorpions in the arenas every afternoon by 5 p.m. and the arenas were then placed in 76.2 cm × 76.2 cm × 63.5 cm cardboard boxes with an infrared video camera that hung over the center of the arena. We used an almost identical arena design as described by Vinnedge and Gaffin, including the Defender 8-channel infrared security camera system (model #HDCB1) [69]. The entire setup was then covered with a 3.65 m × 6.09 m plastic tarp (Blue Hawk Item# 0187739) to ensure that no extraneous light was visible to the scorpions. The internal arena lights were on a timer and remained on from 5:00–5:30 p.m., then turned off from 5:30 p.m.–7:30 a.m., and came back on from 7:30–8:00 a.m. The scorpions’ movements were recorded using the Defender infrared cameras continuously from 5:00 p.m.–8:00 a.m. each observation day. The arenas were then removed from the cardboard boxes and the scorpions were tested behaviorally each morning between 8:00–8:15 a.m. 

### 5.6. Simulated Predatory Encounter

In order to test scorpion’s defensive behaviors, we had to first locate the scorpion. If refuges were present, we removed the refuge the scorpion was under or the refuge it was closest to but left all other refuges unmoved. We elicited defensive behaviors by prodding the scorpion in a fixed pattern for 90 s or until the scorpion fled and hid in a refuge, whichever came first. Scorpions were prodded with the wooden end of a single-tip cotton tip swab (Grainger, Item# 36LF65). The pattern consisted of 10 prods (one prod per second standardized to the beat of a metronome) to the chelae; then 10 prods to one of the last two segments on the posterior side of the metasoma (excluding the telson); and then 10 prods to the carapace of the prosoma just posterior to the median eyes. We repeated this pattern for three cycles so that each body part was touched a total of 30 times and always in the same order, beginning with the chelae and ending with the prosoma. Two researchers were involved in each trial, the same researcher (C.N.H.) always prodded the scorpions, while the other researcher (D.R.N. or J.B.H.) video recorded the behaviors using an iPhone 8 (Apple Inc., Cupertino, CA, USA). 

### 5.7. Data Abstraction

#### 5.7.1. Recordings of Overnight Movements

Each scorpion was allowed to acclimate in the arenas overnight and all movements were recorded for ~15 h per day tested. The recordings (shot at 15 fps using the infrared Defender cameras) were stitched together and sped up 15× their original speed using ShotCut (version 20.04.12), which resulted in a final video at 1 fps. We then exported these files as a MP4s and used a MATLAB (model: R2016a, version: 9.0.0.341360) script [69], supplied by Douglas Gaffin, to perform a frame subtraction analysis for each frame of the exported video. The MATLAB script applied a 200 × 200 grid over the observation area and produced a data file with the X/Y coordinates of the scorpion’s position in each frame. Each square of the applied grid was 0.318 cm^2^. We used these data to calculate the average movement between frames for each trial. We were unable to analyze two of the overnight acclimation videos; therefore, 50 of the 52 videos recorded were analyzed as described.

#### 5.7.2. Recordings of Defensive Behaviors

We analyzed the recordings of defensive behaviors for sting frequency, pinch frequency, venom use, flee duration, latency to sting, latency to pinch, and latency to flee.

Unfortunately, we did not measure the length of each scorpion during the experiment; however, to understand the potential role of sex more clearly, it became obvious that we needed to control for the size of each scorpion. Therefore, we measured the length of the combined prosoma and mesosoma (here after referred to simply as “length”) a posteriori from the recording of defensive behaviors. We calibrated the length measurements based on the known width (2.22 mm) of the probes used to elicit defensive behaviors. We exported a frame from each video that most closely matched the following criteria: scorpion’s body was positioned so that its length was perpendicular to the camera, the probe was nearly touching the scorpion’s body, and both the probe and the scorpion were in focus. We did not correct for special distortion. We then used ImageJ version 1.53a [70] to estimate the scorpion’s length from each image. To increase accuracy, we measured most scorpions three times and used their average length in our statistical models described below. 

### 5.8. Statistical Methods

#### 5.8.1. General Analysis and Order of Defensive Behaviors

The relationship between scorpion length and sex was examined using a Student’s *t* test and the first defensive behaviors used by the scorpions in each trial were analyzed using one-way chi-square. The relationship between mean movement during the acclimation period and the number of refuges and sex was examined using a LMM. This model included mean movement as the dependent variable and refuge number and sex as fixed factors. The interaction between refuge number and sex was also included in the model.

#### 5.8.2. Behavioral Analysis Using Linear and Generalized Liner Mixed Models

We modeled each of the behavioral variables using either LMMs or GLMMs. Because sting frequency, venom use observations, and pinch frequency made use of count data, we used GLMMs with a Poisson distribution (with a log-link function). For the other three behavioral variables (flee duration, latency to sting, latency to pinch, and latency to flee), we analyzed using LMM. 

We used four models to examine the effect of cycle (independent variable) on sting frequency, venom use, pinch frequency, and flee duration (dependent variables). We utilized one model for each dependent variable and only included cycle as the single fixed factor in each. 

We used four separate models to examine the effect of prod location, number of refuges, and sex (independent variables) on sting frequency, pinch frequency, venom use, and flee duration (dependent variables). Here, we again used one model for each dependent variable. To reduce model complexity and facilitate model convergence we omitted cycle from these models. These variables were summed across all cycles prior to model fitting. These models included the number of refuges, prod location, and sex as fixed factors, in addition to mean movement during the acclimation phase, length of each scorpion, and observation time as covariate controls. Since the first cycle was the first time the scorpions were introduced to the stimulus, we removed the observations in the first cycle from these models in order to prevent possible confounding due to the novelty of the stimulus.

We used three additional models to examine the relationship between prod location, number of refuges, and sex (independent variables) on latency to sting, latency to pinch, and latency to flee (dependent variables) again using one model per dependent variable. These models included most of the independent variables described above. However, because each cycle, prod location, and observation time would not influence the absolute time it takes to perform a particular behavior, we omitted them from these models. From these models, we also omitted the observations where the scorpion both did not display the relevant behavior and was not observed for the full trial duration. We assigned 90 s to individuals that were observed for the full trial and did not display the behavior. 

Preliminary analysis (not reported) showed that the only significant interaction for any model we examined was refuge number by sex. Except for those models involving cycle, this interaction was subsequently included in all the models reported in this publication to ensure consistency. For all models, we included the unique ID for each scorpion and the treatment order as random effects to account for the repeated measures design and the potential variation based on the order of treatment.

#### 5.8.3. General Statistical Conventions and Statistical Software

The Student’s *t* test, LMMs, and GLMMs reported here were conducted using Jamovi version 1.2 [71] and the GAMj library [72]. The GAMj library implements post-hoc tests based on the R package emmeans [73]. The GAMj library also calculated marginal and conditional R^2^ for mixed models that we reported in Table 2 and Table 3. Marginal R^2^ measures the variance explained by fixed factors and conditional R^2^ measures the variance explained by both fixed and random factors [74]. We used uncorrected post hoc tests in order to maintain statistical power [75]. One-way chi-square analysis was performed using R version 3.6.1 [76]. We assessed statistical significance using an alpha of 0.05.

## Figures and Tables

**Figure 1 toxins-12-00534-f001:**
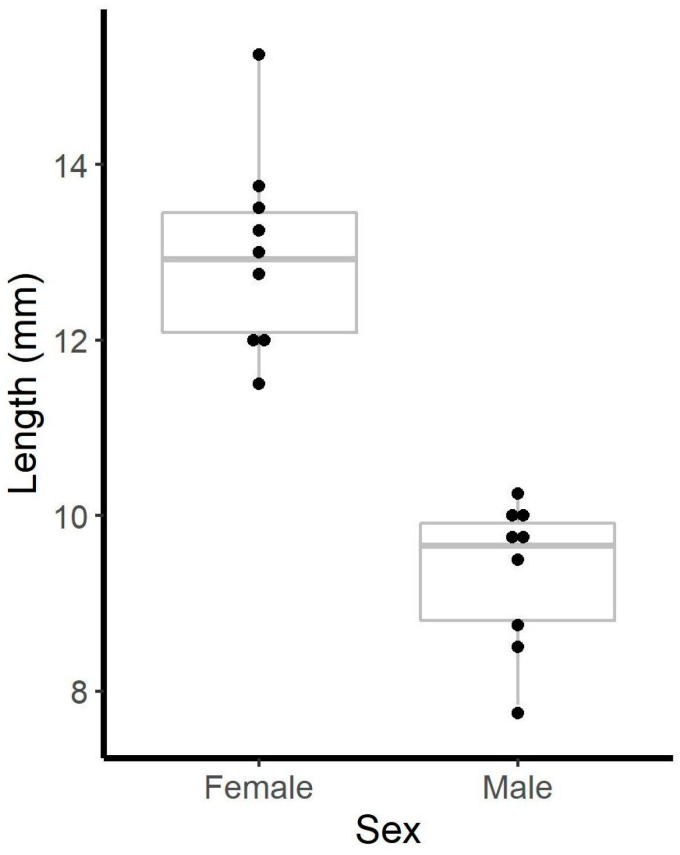
Boxplots showing length (prosoma + mesosoma) distributions of adult male and female *Vaejovis carolinianus*. Dots show underlying data.

**Figure 2 toxins-12-00534-f002:**
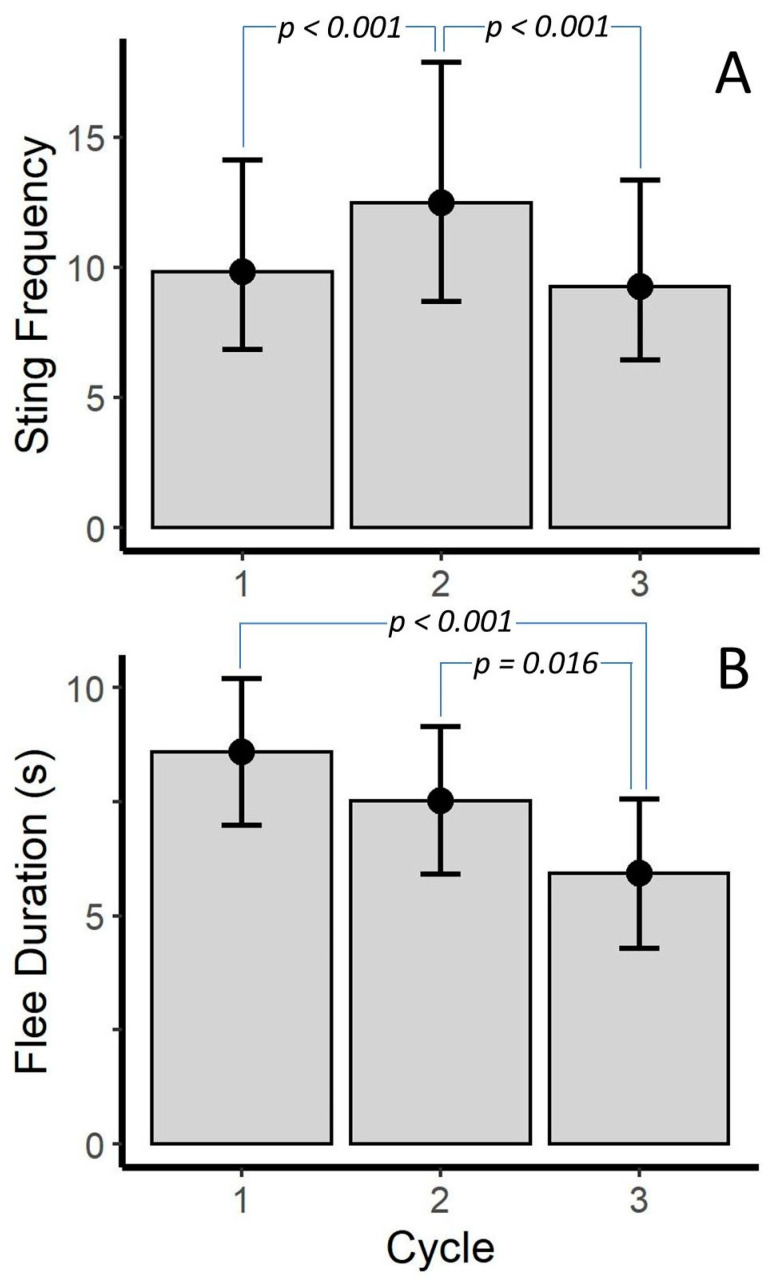
Differences in (**A**) sting frequency (stings per 30 s interval) and (**B**) flee duration across prod cycle for *Vaejovis carolinianus*. Bars represent marginal means from a Poisson Generalized Linear Mixed Model (**A**) and a Linear Mixed Model (**B**). *p*-values are from uncorrected post-hoc tests. Error bars are 95% CI.

**Figure 3 toxins-12-00534-f003:**
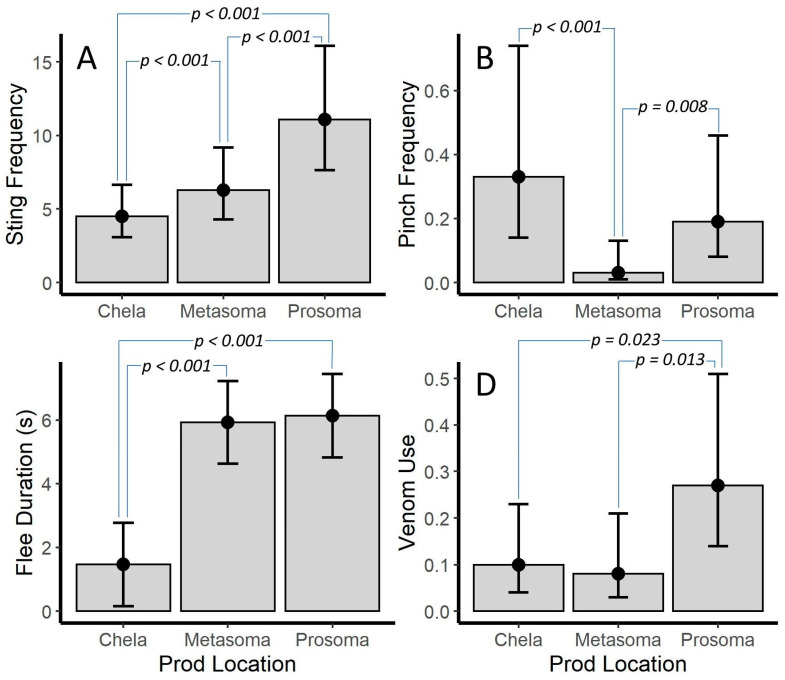
Differences in (**A**) sting frequency (stings per 20 s interval, (**B**) pinches frequency (pinches per 20 s interval), (**C**) flee duration, and (**D**) venom use between stimulation of three body locations for *Vaejovis carolineanus.* Bars represent marginal means from Poisson Generalized Linear Mixed Models (**A**,**B**,**D**) and a Linear Mixed Model (**C**). *p*-values are from uncorrected post-hoc tests. Error bars are 95% CI.

**Figure 4 toxins-12-00534-f004:**
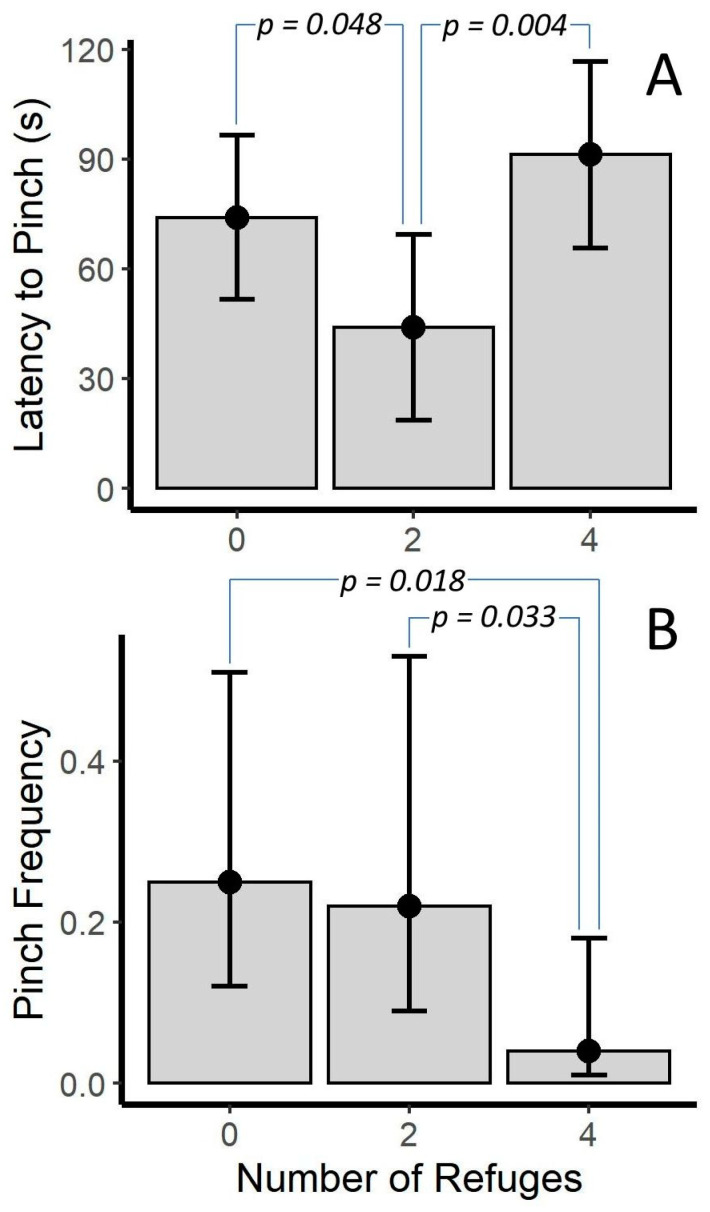
Differences in (**A**) latency to pinch and (**B**) pinch frequency (pinches per 20 s interval) across number of refuges for *Vaejovis carolineanus*. Bars represent marginal means from a Linear Mixed Model (**A**) and a Poisson Generalized Linear Mixed Model (**B**). *p*-values are from uncorrected post-hoc tests. Error bars are 95% CI.

**Figure 5 toxins-12-00534-f005:**
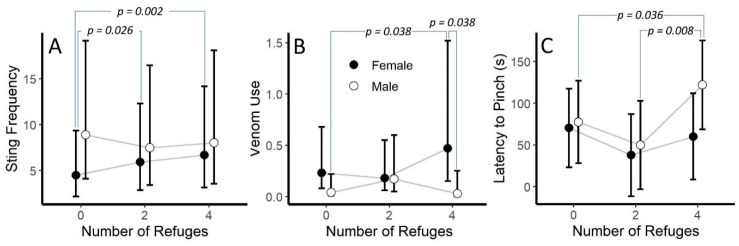
Depiction of the interaction between number of refuges and sex for (**A**) sting frequency (stings per 20 s interval), (**B**) venom use, and (**C**) latency to pinch for *Vaejovis carolineanus*. Bars represent marginal means from Poisson Generalized Linear Mixed Models (A and B) and Linear Mixed Model (**C**). *p*-values are from uncorrected post-hoc tests. Error bars are 95% CI.

**Figure 6 toxins-12-00534-f006:**
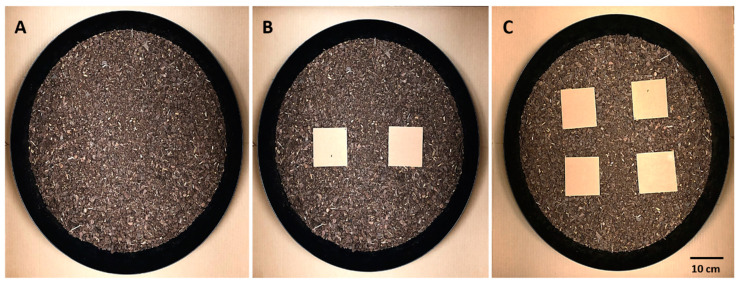
Testing arenas for *Vaejovis carolinianus* risk assessment trails: (**A**) no, (**B**) two, and (**C**) four refuge condition.

**Table 1 toxins-12-00534-t001:** List of terminology and corresponding definitions used to quantify defensive. Responses to a simulated threat.

Terminology	Definitions
Sting	Metasoma movement directed at the stimulus that included the extension of the last metasoma segment so that the telson struck the target, or if the target had moved, where the target had been.
Venom Use	A droplet of venom observed at the tip of the telson. If the droplet grew larger during a successive sting, then this was counted as a new instance of venom use.
Pinch	Open and closure of a chela or chelae around the stimulus.
Flee *	Movement in response to a prod from the stimulus. Fleeing events were measured from the first sign of limb movement until the scorpion became and remained motionless for several frames.
Cycle	Unit of experimental stimulus used to elicit defensive behavior. Each cycle consisted of 10 brief prods to the chelae, 10 prods to the metasoma, and then 10 prods to the prosoma. Each cycle lasted 30 s and always followed the same order: chelae, metasoma, then prosoma.
Mean movement	The average speed of the scorpion as it moved around the arena during overnight acclimation.

* By using the term “flee” we are not implying the scorpion’s intentionality; rather, we selected this term because the concept of “flee” is a more easily understood as a descriptor of this behavior than a more generic term like “movement”.

**Table 2 toxins-12-00534-t002:** Results of omnibus tests for mixed models for the effect of experimental cycle.

	Sting Frequency ^1^		Pinch Frequency ^1^		Venom Use ^1^		Flee Duration ^2^	
	*χ*^2^(*df*)	*p*	*χ*^2^(*df*)	*p*	*χ*^2^(*df*)	*p*	*F*(*df*)	*p*
**Cycle**	31.56(2)	<0.001	1.58(2)	0.455	2.43(2)	0.296	112.94(2)	<0.001
**R^2^ Marginal**	0.03		0.01		0.02		0.07	
**R^2^ Conditional**	0.86		0.36		0.34		0.48	

^1^ Poisson Generalized Linear Mixed Model. ^2^ Linear Mixed Model.

**Table 3 toxins-12-00534-t003:** Results of omnibus tests of fixed effects for mixed models.

	Sting Frequency ^1^	Pinch Frequency ^1^			Venom Use ^1^		Flee Duration ^2^		Latency to Sting ^2^		Latency to Pinch ^2^		Latency to Flee ^2^	
	*χ*^2^(*df*)	*p*	*χ*^2^(*df*)	*p*	*χ*^2^(*df*)	*p*	*F*(*df*)	*p*	*F*(*df*)	*p*	*F*(*df*)	*p*	*F*(*df*)	*p*
**Prod Location**	167.27(2)	<0.001	14.74(2)	<0.001	9.03(2)	0.011	25.36(2)	<0.001	-	-	-	-	-	-
**# of Refuges**	2.13(2)	0.344	5.63(2)	0.06	1.56(2)	0.459	0.55(2)	0.578	1.00(2)	0.38	7.04(2)	0.007	0.38(2)	0.687
**Sex**	0.29(1)	0.588	0.01(1)	0.909	2.88(1)	0.09	0.29(1)	0.596	0.25(1)	0.622	0.44(1)	0.516	2.10(1)	0.115
**Refuges × Sex**	11.66(2)	0.003	1.88(2)	0.39	6.07(2)	0.048	1.54(2)	0.218	0.52(2)	0.602	3.04(2)	0.073	0.49(2)	0.617
**Movement**	17.64(1)	<0.001	1.09(1)	0.297	6.84(1)	0.009	0.07(1)	0.797	5.41(1)	0.026	0.47(1)	0.499	0.50(1)	0.486
**Length**	0.63(1)	0.426	0.10(1)	0.749	0.01(1)	0.927	0.09(1)	0.764	1.00(1)	0.333	0.02(1)	0.881	2.04(1)	0.161
**Time Observed**	0.67(1)	0.414	2.49(1)	0.114	0.41(1)	0.521	0.59(1)	0.218	-	-	-	-	-	-
**R^2^ Marginal**	0.36	0.5			0.46		0.27		0.16		0.25		0.07	
**R^2^ Conditional**	0.84	0.59			0.47		0.39		0.44		0.68		0.08	

^1^ Poisson Generalized Linear Mixed Model. ^2^ Linear Mixed Model.

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
