# Peer review of "Risk Assessment and the Effects of Refuge Availability on the Defensive Behaviors of the Southern Unstriped Scorpion (*Vaejovis carolinianus*)"

_toxins, 2020, doi:10.3390/toxins12090534_

Round 1

Reviewer 1 Report

This study presents an investigation into information gathering and risk assessment in scorpions. The manuscript is well written and presents a new look into environmental awareness in arthropods. I have a few questions about the methodology that were not explained in sufficient detail, specifically regarding the shelters.

Please explain the structure of the shelters in more detail. I’m not an expert in this field, but I could imagine that the scorpions may have seen the structures more as a stressor (novel object) than as a shelter. This would in part explain the unexpected results.

Were the shelters removed before the experiment? If not, did scorpions never retreat into them when prodded? This would certainly indicate that the shelters were not performing their intended duty…

Also, I strongly encourage the authors to include body size as a factor in their models. Again, I’m not an expert in arthropods, but larger individuals may act less aggressive because they feel less threatened in general due to their large size. Alternatively, they may represent more aggressive behavioural types and only became larger because they are better at gaining resources through high aggression and/or bold behaviour. Even if size was not specifically measured, you have ample video material with size standards (i.e., arena, shelters) to estimate size a posteriori.

Specific comments:

26 – 40: This paragraph could probably handle a few more references.

431: Give more detail about the shelter structure. Is it a box with two entrances?

436: Please specify more clearly, how this was done. Were all or a subset of individuals tested simultaneously or was each individual tested (for three days) and then the next (meaning the entire trial was over 50 days long)?

450: tested

177 – 180: Did you test for an interaction of movement with refuges? I could imagine that, for instance, an individual may move less in the presence of shelter. In general, movement is certainly not independent of shelter

231: to be able

270: This is a very lazy way of going about this. You could easily re-run all models with body size included (see main comments).

279 – 299: Again, movement could have been affected by shelter availability, as may have defensive or aggressive behaviours. For instance, in a situation with lots of shelter, an individual may move less (because it spends more time hidden), while at the same time being less aggressive (because it knows that it can quickly use a nearby shelter if the situation becomes too threatening) or more aggressive (because it defends its shelter), and vice versa. So, the two measures are not independent.

300 – 305: Yes, good point. In general, you may find that certain syndromes (e.g., activity – aggression) may prevail across treatment conditions (e.g., shelter availability or prodded body region) and potentially even override any effect of these factors. Because of the unpredictability of behaviour, it is typically advisable to have relatively large sample sizes in these types of studies…

Author Response

Responses to Reviewer 1, Manuscript submission “toxins-881256”

Point 1: Please explain the structure of the shelters in more detail. I’m not an expert in this field, but I could imagine that the scorpions may have seen the structures more as a stressor (novel object) than as a shelter. This would in part explain the unexpected results.

            We included an additional description of the refuges and rationale for why we designed the refuges the way we did: see lines 476-482. This species of scorpion is commonly found under bark and rocks. Thus, we believe that having large objects within their habitat that they can climb over and under mimics the conditions this species experience naturally. Scorpions were also able to acclimate to the mock habitats for several hours prior to behavioral testing. 

Point 2: Were the shelters removed before the experiment? If not, did scorpions never retreat into them when prodded? This would certainly indicate that the shelters were not performing their intended duty…

            Section 5.6 describes how simulated predatory encounters were performed. Lines 506-508 specifically addresses how refuges influenced the testing procedure. To summarize, if a refuge was present, we removed the refuge the scorpion was closest to before behavioral testing started. This was either the refuge the scorpion was hiding under or the closest to the scorpion in a straight line. Behavioral testing lasted either 1 minute 30 seconds or until the scorpion ran under another refuge. We added the phrase “whichever came first” to emphasize the role refuges may have played in behavioral tests. We have also updated the results to include how many individuals fled to a refuge during testing, see lines 144-145.

Point 3: Also, I strongly encourage the authors to include body size as a factor in their models. Again, I’m not an expert in arthropods, but larger individuals may act less aggressive because they feel less threatened in general due to their large size. Alternatively, they may represent more aggressive behavioural types and only became larger because they are better at gaining resources through high aggression and/or bold behaviour. Even if size was not specifically measured, you have ample video material with size standards (i.e., arena, shelters) to estimate size a posteriori.

            We followed the reviewer’s suggestion and estimated the “length” of each scorpion from images taken from the recordings of defensive behaviors. As a result of this change, we have updated all relevant statistical models, results, figures, tables, and the discussion and method sections. We acknowledge the usefulness of this suggestions and appreciate the advice the reviewer gave.

Specific comments:

26 – 40: This paragraph could probably handle a few more references.

            We have added several additional citations to provide support for the concepts of optimal behavior (line 30), its role in survival (line 34), and the definition of risk assessment (lines 35-36).

431: Give more detail about the shelter structure. Is it a box with two entrances?

            See our response to point 2 above.

436: Please specify more clearly, how this was done. Were all or a subset of individuals tested simultaneously or was each individual tested (for three days) and then the next (meaning the entire trial was over 50 days long)?

            We added a new section (5.4) to clarify how we standardized and randomized our experiment to mitigate potential confounding variables. We hope this helps to add clarity to our experimental design. See lines 485-491. 

450: tested

            We fixed this typo. Thank you for catching it.

177 – 180: Did you test for an interaction of movement with refuges? I could imagine that, for instance, an individual may move less in the presence of shelter. In general, movement is certainly not independent of shelter

            We updated the results section to include results for refuge number x mean overnight movements and sex by mean overnight movement. Neither were significant, see lines 126-130.

231: to be able

            This typo has been found and fixed.

270: This is a very lazy way of going about this. You could easily re-run all models with body size included (see main comments).

            See our response to Point 3 above.

279 – 299: Again, movement could have been affected by shelter availability, as may have defensive or aggressive behaviours. For instance, in a situation with lots of shelter, an individual may move less (because it spends more time hidden), while at the same time being less aggressive (because it knows that it can quickly use a nearby shelter if the situation becomes too threatening) or more aggressive (because it defends its shelter), and vice versa. So, the two measures are not independent.

            We agree that this may be true in some cases and should be tested for. Our results did not find an interaction between the number of refuges or sex and how scorpions moved overnight. In addition, we also included mean overnight movement as a covariate in our models. Thus, we feel design and analysis accounted for this possibility. However, further research is necessary to confirm our results. 

300 – 305: Yes, good point. In general, you may find that certain syndromes (e.g., activity – aggression) may prevail across treatment conditions (e.g., shelter availability or prodded body region) and potentially even override any effect of these factors. Because of the unpredictability of behaviour, it is typically advisable to have relatively large sample sizes in these types of studies…

            We agree with this suggestion. We believe our results are preliminary as they represent a first attempt to investigate how features in the environment may influence defensive venom use. This was our first time collecting and working with this species of scorpions and we were not able to find as many individuals as we would have liked. We have since become much better at finding this species and our future studies will have more robust samples sizes to ensure we have good statistical power.

Reviewer 2 Report

Dear Authors,

I applaud you for performing an extremely sound study and for a well written manuscript that was a joy to read. I have only minor comments, which I included in the attached pdf. Some of those I consider more important and are listed below. Please consider all comments as suggestions only, none are obligatory, and use them by your judgement.

Sample size in the abstract does not match the one you report in other parts of the manuscript. Please check again all parts and specify the right sample size consistently.

Given the "methods last" structure of the manuscript, you should pay more attention that the content presented in results is crystal clear without having to jump down and consult the methods. I though "cycle" and some other stuff was unclear without checking the methods. I did not mark all. If this demands too much rewriting, I suggest putting methods before the results (if this is in line with the journal format).

Figure 1B hints that some habituation to the stimulus could have occurred. Do you think so too? If yes, maybe give it a quick mention in results or discussion.

Figure 4 lacks an explanation of which circle color was used for males and females. Please add this information.

Personally, I found the discussion quite wordy and would prefer if some ideas were presented in a more concise way. But I do not want to interfere with your writing style. It is up to you if you want to shorten some parts.

I found the text on Statistical Methods hard to follow because I think it is presented in a different order than the results are. I suggest unifying this by reorganizing the text a bit. I also suggest adding some details regarding the analyses such as specifying which method was used for post-hoc tests and which for R2 calculation. A few citations in this section are wrongly formatted and missing from the reference list.

Please reference you supplementary material files in the main text in appropriate sections. Without a reference/link it is not clear why they are important for the coherence of the manuscript.

Good luck in your future scorpion endeavours and all the best,

Author Response

Responses to reviewer 2, manuscript “toxins-881256”

Point 1: Sample size in the abstract does not match the one you report in other parts of the manuscript. Please check again all parts and specify the right sample size consistently.

            The abstract has a 200-word limit, and we are currently at 199-words. We did test 18 total scorpions; however, 17 of these were tested fully and one was tested only one of the three times. Therefore, some statistical tests have N = 18 and some N = 17. Because of the word limit, we felt it was simpler to use the sample size we provided and explain the nuance more fully in the article itself. We hope that the reviewer and future readers of the article understand this discrepancy. But, because of the word limit, we will leave the sample size reported in the abstract unchanged. To try and add clarity for the reader we have added wording about differences in sample size both in the results (see line 123-124) and in the methods (462-464).

Point 2: Given the "methods last" structure of the manuscript, you should pay more attention that the content presented in results is crystal clear without having to jump down and consult the methods. I though "cycle" and some other stuff was unclear without checking the methods. I did not mark all. If this demands too much rewriting, I suggest putting methods before the results (if this is in line with the journal format).

            We do apologize for the lack of clarity. This was our first-time writing using the methods last format and recognize that we needed to make changes to our writing style to compensate for this format. We have rewritten the results section to include the necessary descriptions of key terminology early in the results, including a table of terms and definitions (see table 1). We hope these changes allow readers to easily understand the experiment so that they can read the results without having to jump down to the methods first. 

Point 3: Figure 1B hints that some habituation to the stimulus could have occurred. Do you think so too? If yes, maybe give it a quick mention in results or discussion.

            We agree that this is a possible interpretation for the decrease in response to our simulated attacks across cycle. However, we did not specifically test for habituation and cannot rule out fatigue as a possible explanation. We have updated the manuscript to include both of these possible interpretations, see line 392-394.

Point 4: Figure 4 lacks an explanation of which circle color was used for males and females. Please add this information.

            We have updated the figure to include the necessary legend.

Point 5: Personally, I found the discussion quite wordy and would prefer if some ideas were presented in a more concise way. But I do not want to interfere with your writing style. It is up to you if you want to shorten some parts.

            We have edited the manuscript extensively to incorporate reviewer comments and have also tried to reduce the wordiness in the process. However, doubtless, we could continue to refine our writing in this manuscript. Due to the time constraints of responding to the reviews and the importance of other comments requiring substantial changes to the manuscript we have done the best we can regarding this comment.

Point 6: I found the text on Statistical Methods hard to follow because I think it is presented in a different order than the results are. I suggest unifying this by reorganizing the text a bit. I also suggest adding some details regarding the analyses such as specifying which method was used for post-hoc tests and which for R2 calculation. A few citations in this section are wrongly formatted and missing from the reference list.

We have reorganized the section on statistical methods in order to more closely match the order of the results. Wording and citations have also been added to explain the methods used for the post-hoc tests and for the R2 calculations, see section 5.8.3.

Point 7: Please reference you supplementary material files in the main text in appropriate sections. Without a reference/link it is not clear why they are important for the coherence of the manuscript.

            We have updated all figure and tables, renumbered some, and made sure to reference each in the body of the manuscript. Thank you for this comment.

Other comments from attached PDF:

Lines 31-33: This sounds as a common definition of risk assessment. I suggest to provide a citation to support it.

            We searched for a formal definition of risk assessment but were unable to find this term clearly defined. Thus, we wrote this general description so that the term would be clear to those not familiar with the concept. Since writing this description, another researcher suggested we also search for the term “threat assessment” and in doing so we found the following a relevant article by Blanchard et al. 2011. We have included this as a supporting reference for the description in question.

Line 80: Replace with: metering

            The typo has been corrected as suggested

Lines 82-83: This is practically always the case in any field of science. Consider rewriting to something like: “However, several knowledge gaps exist”

            We agree with this critique and we have changed the sentence similar to what was suggested, see lines 85-86.

Line 86: There is always more work that can be done. Consider rewriting to something like: ”Other families are thus understudied.”

            We followed the reviewer’s suggestion and modified our language to emphasize the lack of work in most scorpion families: see line 89.

Table 2: correct movement duration to Flee duration to be consistent with text.

            We did this and replaced the table with the updated text.

Update figure 4 to include a legend for shapes and color of circles

            We apologize for this oversight and updated the figure to include the necessary legend.

Line 180: Maybe add a less technical summary. Scorpions with higher levels of base activity displayed aggressive defensive behaviors more often.

            We have updated the results section to more clearly explain our experiment and included a table with definitions of key terms. We hope that this will clarify these terms. However, for the sake of consistency, we have continued to use the phrases “mean movement during overnight acclimation” or mean overnight movement” when referring to this variable. We believe that consistency will also help to make the text more readable.

Line 186: Not clear what is “others.”

            We changed “others” to “other body parts” to eliminate confusion: see lines 224-225.

Lines 240-245: This sounds as better fitted for the results section (descriptive part) and not for discussion.

            We agree with this critique and have updated the results section to include general results, see lines 121-163.

Line 399: Your sample size was relatively low (18) and behavioral traits are often very variable. Maybe discuss the need for greater sample size to get a reliable signal from a very variable and labile trait.

            We agree that our results need additional testing, particularly the results concerning the role refuges may have played. We have softened the language here to emphasize this point: see lines 411-413.

Lines 400-401: I think this is probably not true. At least for animals in general. In example, we have published a paper where we showed that cave environment caused that some cave isopods use refuges less than surface isopods. In case you are interested see below. Maybe I misunderstood your message. In that case, please rephrase the sentence and be more specific.

Fišer, Ž., Prevorčnik, S., Lozej, N., & Trontelj, P. (2019). No need to hide in caves: shelter-seeking behavior of surface and cave ecomorphs of Asellus aquaticus (Isopoda: Crustacea). Zoology, 134, 58–65.

            You are right, we were not clear enough here. We have revised the sentence to represent the “newness” of our study more specifically: see lines 441 & 442. Also, thank you for the suggested article. We were not aware of this study and look forward to learning from it.

Line 426: Does it need to be so precise? I would write 24.

            We measured temperature in Fahrenheit and converted it to Celsius, leading to this decimal. We have decided to leave the temperature description as is. For those of us that work and think in Fahrenheit the decimal will provide more accurate information.

Line 431: Change figure caption to: Testing arenas for Vaejovis carolinianus risk assesment trails: A) zero, B) two, and C) four refuge condition.

I also suggest to add a scale bar to A, B, or C. It is easier to get the size than from the text.

            We have updated the figure to reflect the recommended changes.

Line 435: It is not clear to me whether they were lifted a bit, producing a narrow space between the bottom and the shelter? Or such lifting is not neccessary for scorpions as they burrow underneath any way, and refuges were put flat on the ground. I am not a scorpion expert, sorry.

            We have rewritten this section to add more information about refuge construction and use: see lines 476-482 We believe this should clear up your concerns.

Line 466: Please mention the time resolution (frames per second) you used to record the original video and fps used to calculate movement variables if this was different.

            This information was added to Section 5.7.1.

Reviewer 3 Report

This is an interesting manuscript presenting data worth publishing. However, I think that data analysis and the presentation need some work before it is possible to assess the manuscript. It is not fully clear how you did the experiments and how you analysed them. For example, it is not clear how you repeated the experiments with no, 2 and 4 refuges. Did you use a random order each time or a fixed order? Anyways, this makes the trials dependent of each other so the order is important to consider unless it was random. There is also the same problem for body parts, as you always first prodded the chelae, then the metasoma and finally the prosoma, which is the same order as the effect (highest effect on prosoma), so actually it might be a cumulative effect. Did you make sure to assess the combined effect of body part and experimental order?

Generally, there is an extreme overuse of significance testing. This is especially a problem because the study design makes trials highly autocorrelated, so there is a serious pseudoreplication issue here. Some researchers (e.g. Hurlbert SH. 1984. Pseudoreplication and the design of ecological field experiments. Ecological Monographs 54, 187-211) would even see the use of one arena as pseudoreplication, but I think the important sample size is the number of individuals. Confidence intervals (for the variation among individuals) are sufficient to illustrate differences. I therefore recommend to remove all significance testing from the manuscript, which is anyways not much of significance (see e.g. Johnson, D.H. 1999. The insignificance of statistical significance testing. Journal of Wildlife Management 63: 763-772.).

There is also a large amount of superfluous information throughout the manuscript. For example, in the methods you write that you repeated experiments 4 times (cycles) but you only present 3 cycles. As the 4th cycle did not influence the first 3 and you could anyways not use the results, why writing about them at all? Please avoid providing superfluous information, it distracts the reader from the essential parts of the manuscript. You report that you recorded overnight movements but I could not find any results for this (except the pictures of movements in the supplement but no anaylsis). Why did you mention it here? The naming of the trials as 2 and 4 refuge conditions is misleading. As I understand it, you always removed one refuge, so actually the scorpions had 1 or 3 refuges for retrieval. This might also have led to a partly unknown environment for the scorpions.

Some parts of the abstract are unclear as not enough background information is available. The reader must be able to understand the abstract before reading the paper. As the method section is at the end of the manuscript, you also need to better explain what you analysed in the result section as you cannot expect the reader to go first to the end of the manuscript to be able to understand what comes before. The introduction is too detailed, it is more important to set up the objectives of the study than to provide a complete state of the knowledge.

You should also edit the writing for clarity, I provided some examples in the detailed comments, but the language would benefit from editing throughout the manuscript. Please write clear and unambiguously, you should also avoid the passive voice wherever possible.

Detailed comments:

Line 7-9. Sentence unclear, please reword. Do you mean “a few studies with a few species/from one genus/family(?) have found…”?

Line 8-9. “both release and volume expelled”, unclear, please specify

Line 9. What do you mean by “passive features”?

Line 10 change “N = 9 females and males” to “9 females and 9 males”.

Line 13. By “distal metasoma” do you mean directly touching the telson? Maybe better write the distal end of the metasoma.

Line 18 what is the “four-refuge condition”?

Line 26. What do you understand as “internal information”?

Line 31. The sentence would benefit from a citation. And the next.

Line 34. Awkward sentence as it is not the decision that incur risk.

Line 49. Awkward sentence, what do you mean by established? Studied?

Line 120. Probably “no-refuge” is better to use than “zero-refuge”. Or you could just write here: “when no refuge was available, fleeing…”. What are the 3 refuge conditions? If the methods are at the end of the paper, then you have to explain this here.

Line 121. The test results have no context here. What did you compare?

Line 122. You need to define here “four-hide” (also in the abstract).

Line 124-125. Not understandable why the two-refuge condition was unique. Was there more than 1 scorpion in the arena? Why would you remove scorpions from analysis only because they responded with fleeing and stinging?

Line 118-128. It would be much easier to grasp the results if you presented all these results in a table.

Line 131. First sentence superfluous. Start directly with the second. No sense to repeat here all p values. Completely unclear what you understand here as a cycle.

Line 428. You mention here that you fed the individuals once per week but you do not specify when you performed the experiments in relation to the moment of feeding. Were all tests done at a specific time after feeding? Or is this another variable to consider?

Line 436. Not clear what you mean by “Each scorpion was tested three times in a randomized order” do you mean that the order of the experiments with various numbers of refuges was randomized, or that the order of individuals in the trials was random? Also, not clear if the cards were laying flat on the flour substrate or if they had a certain height.

Table 1. Little informative as only test results. It is more important to provide effect size, e.g. Table 1 is much more important as result (although seems to be a repetition of figures, just decide if you present data as figure or table, not both). If you replace here SD with CI, then you also do not need any significance testing here. However, you need to provide (here and elsewhere) units. For example, what does sting number mean (number of stings per (which?) interval)? Be more precise in information and less in precision of digits (e.g. if you count number of stings in full numbers, there is no sense to provide means with 2 decimals. Maximal 1 is useful). “Venom use number” is awkward, you mean “times of venom use”? Latency needs a unit and reference point (is it seconds after first/last stimulation?)

Figure 1. Not clear what exactly is sting frequency (in relation to which time interval). Is flee duration a correct term? Not “flight/escape duration”? Do you mean by this the duration from first movement to time again immobile? Why would this be important? Is it not better to use e.g. the distance moved? When you present CI, you do not need to provide p values, the CIs are more informative. What is here the same size, 18 individuals?

Table 2. This table is of little value as you only report test results but no effect sizes, can be deleted.

Figure 2. Frequency needs a time interval. P values useless as they are even significant when CI largely overlap. CIs are more conservative. General problem here: Sting frequency increases with the order of trials, you need to prove here that the reactions were consistent over the 3 cycles to show there is no cumulative effect. Abbreviation for second is s not sec.

Figure 3. Difficult to assess the results without knowing the order of experiments.

Figure 4. Did you use here marginal means and not arithmetic means? Why? The number of stings for example can easily be calculated as a mean for the 18 individuals. It does not look that there is a difference between sexes as all CIs overlap. It is also not possible to know which sex has which symbol.

Figure S1. I fail to see the point of this figure, what do you want to show by these plots?

Author Response

Responses to reviewer 3, manuscript “toxins-881256”

Point 1: For example, it is not clear how you repeated the experiments with no, 2 and 4 refuges. Did you use a random order each time or a fixed order? Anyways, this makes the trials dependent of each other so the order is important to consider unless it was random.

The order of treatment was randomized with respect to the number of refuges. We have added language to make this clearer (lines 124-125 and 485-486).

Point 2: There is also the same problem for body parts, as you always first prodded the chelae, then the metasoma and finally the prosoma, which is the same order as the effect (highest effect on prosoma), so actually it might be a cumulative effect. Did you make sure to assess the combined effect of body part and experimental order?

We admit that our study design may not have completely controlled for the cumulative effects you describe. However, we did take steps to minimize confounding. First, during the behavioral trials we repeated the pattern of prodding to the chelae, then metasoma, and then prosoma three times (three cycles). Second, in our analysis, we omitted observations from the first cycle (see lines 565-567). We assumed that scorpions would go from zero to a maximum perception of threat during the first cycle and this maximum perception of threat would be maintained during the second and third cycles. Since the perception of threat cannot increase when it is at maximum, we reasoned that this would negate any confounding cumulative effects due to threat perception.

Our analysis of defensive behaviors between cycles lends some support to our rational in that scorpions tended to sting more in the second cycle. However, our results also suggest the possibility of habituation and a possible reduction of threat perception in at least the third cycle. But this possible reduction should work against a possible cumulative effect. One might argue that seeing a greater intensity of defensive behaviors associated with prodding to the prosoma in spite of a possible decrease in perceived threat between cycles two and three might increase our confidence in the result.

Point 3: Generally, there is an extreme overuse of significance testing. This is especially a problem because the study design makes trials highly autocorrelated, so there is a serious pseudoreplication issue here. Some researchers (e.g. Hurlbert SH. 1984. Pseudoreplication and the design of ecological field experiments. Ecological Monographs 54, 187-211) would even see the use of one arena as pseudoreplication, but I think the important sample size is the number of individuals. Confidence intervals (for the variation among individuals) are sufficient to illustrate differences. I therefore recommend to remove all significance testing from the manuscript, which is anyways not much of significance (see e.g. Johnson, D.H. 1999. The insignificance of statistical significance testing. Journal of Wildlife Management 63: 763-772.).

We are puzzled regarding your suggestion that our design has “a serious pseudoreplication issue.” It is true that there is autocorrelation and lack of independence inherent in our design. Indeed, trials involving the same individual scorpion will lack independence and be autocorrelated. However, such is the nature of repeated measures designs and our method of analysis (GLMM and LMM) should account for this.

In regard to the use of significance testing, we recognize the potential for the misuse of this statistical methodology. The paper you mentioned (Johnson, 1999) effectively describes this potential for misuse. However, Johnson does admit that most of the objections to significance testing stem from their potential misuse rather than their intrinsic value. While Johnson suggests avoiding significance testing and suggests alternatives, others, such as (Murtaugh, P. A. (2014). In defence of P values. Ecology, 95(3), 611–617) suggest that significance testing isn’t any less arbitrary than proposed alternatives, such as confidence intervals, and argues that the “choice of which to use should be stylistic, dictated by details of the application rather than by dogmatic, a priori considerations.” As such, we feel our use of significance testing is largely appropriate. However, for those that desire more information about our analysis, we have followed a suggestion from Johnson (1999) and included two supplemental tables (S1 and S2) that include the estimates of all model coefficients along with their 95% confidence intervals.

Point 4: There is also a large amount of superfluous information throughout the manuscript. For example, in the methods you write that you repeated experiments 4 times (cycles) but you only present 3 cycles. As the 4th cycle did not influence the first 3 and you could anyways not use the results, why writing about them at all? Please avoid providing superfluous information, it distracts the reader from the essential parts of the manuscript.

We have removed all references to the fourth cycle from the updated manuscript.

Point 5: You report that you recorded overnight movements but I could not find any results for this (except the pictures of movements in the supplement but no anaylsis). Why did you mention it here?

We have added analysis of overnight movements to the results (see lines 126-130).

Point 6: The naming of the trials as 2 and 4 refuge conditions is misleading. As I understand it, you always removed one refuge, so actually the scorpions had 1 or 3 refuges for retrieval. This might also have led to a partly unknown environment for the scorpions.

We named the trials based on the number of refuges the scorpions experienced during the overnight acclimation phase of the experiment. This is what the scorpions were familiar with prior to the behavioral testing when a refuge was removed. We agree that the removal of a refuge could have led to a partly unknown environment for the scorpions. However, these scorpions are commonly found under rocks – rocks that must be disturbed in order to expose the scorpions. We feel that eliciting defensive behaviors in the context of a somewhat disturbed habitat may be biologically relevant.

Point 7: Some parts of the abstract are unclear as not enough background information is available. The reader must be able to understand the abstract before reading the paper.

We have revised our abstract for clarity by removing language that is only defined on the body of the manuscript. We hope that the abstract can now stand on its own without the reader needing to look at the body of the manuscript to understand it.

 Point 8: As the method section is at the end of the manuscript, you also need to better explain what you analysed in the result section as you cannot expect the reader to go first to the end of the manuscript to be able to understand what comes before.

We have updated the first part of the results (lines 121-145) to explain enough of the methodology so that a reader can understand the results.

Point 9: The introduction is too detailed, it is more important to set up the objectives of the study than to provide a complete state of the knowledge.

We respectfully disagree. We feel that part of the purpose of the introduction is to explain how a study adds to our knowledge on a subject. This requires at least some detail about the current state of knowledge in this area. In addition, the extra detail makes this article more accessible to readers who have less expertise in this field.

You should also edit the writing for clarity, I provided some examples in the detailed comments, but the language would benefit from editing throughout the manuscript.

We appreciate your help and the help of the other reviewers in this area. We hope that the changes we have made have improved the clarity of our manuscript.

Point 10: Please write clear and unambiguously, you should also avoid the passive voice wherever possible.

We have tried to find and rewrite wording that uses the passive voice.

Detailed comments:

Line 7-9. Sentence unclear, please reword. Do you mean “a few studies with a few species/from one genus/family(?) have found…”?

We have removed this wording.

Line 8-9. “both release and volume expelled”, unclear, please specify

We have changed this wording to read “both when it [venom] is released, and the total volume expelled.” We hope this adds clarity.

Line 9. What do you mean by “passive features”?

We have removed this wording and replaced it with “awareness of environmental features”

Line 10 change “N = 9 females and males” to “9 females and 9 males”.

            We edited the manuscript to incorporate the suggested change.

Line 13. By “distal metasoma” do you mean directly touching the telson? Maybe better write the distal end of the metasoma.

            We removed the word distal to reduce confusion here. This change does not add more clarity but because of the 200 word limit to the abstract, we believe that a more general description will have to suffice.

Line 18 what is the “four-refuge condition”?

            We rephrased this phrase as “when four refuges were present.” We hope this clears up the confusion. 

Line 26. What do you understand as “internal information”?

            We provided several examples of internal information that may influence how an organism responds to a situation. See lines 28-29.

Line 31. The sentence would benefit from a citation. And the next.

            We have added a relevant citation that reviews this topic.

Line 34. Awkward sentence as it is not the decision that incur risk.

            We have rewritten the sentence to reflect the cause and effect relationship more clearly. See lines 37-38.

Line 49. Awkward sentence, what do you mean by established? Studied?

            We have replaced the word “established” with “documented”. See line 51.

Line 120. Probably “no-refuge” is better to use than “zero-refuge”. Or you could just write here: “when no refuge was available, fleeing…”. What are the 3 refuge conditions? If the methods are at the end of the paper, then you have to explain this here.

We have replaced “zero-refuge” with “no-refuge” throughout the manuscript. As stated before, we have also included language at the beginning of the results section (lines 121-122) that explains the 4 refuge conditions.

Line 121. The test results have no context here. What did you compare?

Hopefully, our addition to the first part of the results section explains the needed context.

Line 122. You need to define here “four-hide” (also in the abstract).

We have replaced this wording with “four-refuge”. We have also removed this wording from the abstract.

Line 124-125. Not understandable why the two-refuge condition was unique. Was there more than 1 scorpion in the arena? Why would you remove scorpions from analysis only because they responded with fleeing and stinging?

In the context, we are simply saying that trials involving the two-refuge condition were unique because they were the only ones were scorpions had simultaneous first behaviors. We did not see this in any other refuge condition. We never had more than one scorpion in an arena at one time.

The three scorpions that had simultaneous first behaviors were removed from statistical analysis because the chi-square test assumes independent data. Placing a single scorpion in two categories would violate this assumption.

Line 118-128. It would be much easier to grasp the results if you presented all these results in a table.

We decided against a table in this case because the information presented here is all in support the simple conclusion that fleeing is the most common first defensive behavior.

Line 131. First sentence superfluous. Start directly with the second. No sense to repeat here all p values. Completely unclear what you understand here as a cycle.

We only repeated p-values in both the table and the paragraph if they were significant (p < 0.05). Most p-values in the paragraph are from the post-hoc tests which are not found in the table. Though they are repeated in figure 2.

Line 428. You mention here that you fed the individuals once per week but you do not specify when you performed the experiments in relation to the moment of feeding. Were all tests done at a specific time after feeding? Or is this another variable to consider?

We included the following (line 486-487) in the manuscript “We standardized the scorpions feeding schedule so that each scorpion was fed one week prior to testing.”

Line 436. Not clear what you mean by “Each scorpion was tested three times in a randomized order” do you mean that the order of the experiments with various numbers of refuges was randomized, or that the order of individuals in the trials was random? Also, not clear if the cards were laying flat on the flour substrate or if they had a certain height.

We have updated the methods (lines 485-486) to clarify. We mean that the order the scorpions were exposed to each refuge condition was randomized.

Table 1. Little informative as only test results. It is more important to provide effect size, e.g. Table 1 is much more important as result (although seems to be a repetition of figures, just decide if you present data as figure or table, not both). If you replace here SD with CI, then you also do not need any significance testing here. However, you need to provide (here and elsewhere) units. For example, what does sting number mean (number of stings per (which?) interval)? Be more precise in information and less in precision of digits (e.g. if you count number of stings in full numbers, there is no sense to provide means with 2 decimals. Maximal 1 is useful). “Venom use number” is awkward, you mean “times of venom use”? Latency needs a unit and reference point (is it seconds after first/last stimulation?)

We assume you are referring to the first table in the supplemental material. We opted to remove this table.

Figure 1. Not clear what exactly is sting frequency (in relation to which time interval). Is flee duration a correct term? Not “flight/escape duration”? Do you mean by this the duration from first movement to time again immobile? Why would this be important? Is it not better to use e.g. the distance moved? When you present CI, you do not need to provide p values, the CIs are more informative. What is here the same size, 18 individuals?

We have updated the figure captions to include the appropriate time interval when dealing with sting or pinch frequency. We added table 1 in order to define and clarify our use of the term ‘flee duration.’

In terms of why we measured flee duration, we took our cue from (Martin, P. and Bateson, P. 1995. Measuring Behaviour: An introductory guide. Cambridge University Press.) which states that “States are behavior patterns of relatively long duration, such as prolonged activities, body postures or proximity measures. The salient feature of states is their duration (mean or total duration or the proportion of time spent performing the activity).” We considered fleeing as a state and, therefore, measured its duration. This is not to say that measuring the distance moved during fleeing would not have been informative, it is just something we didn’t choose to measure in this study.

Table 2. This table is of little value as you only report test results but no effect sizes, can be deleted.

As explained above, we feel that p-values, properly understood, can be useful. We provide estimates of model coefficients and associated 95% confidence intervals in tables S1 and S2

Figure 2. Frequency needs a time interval. P values useless as they are even significant when CI largely overlap. CIs are more conservative. General problem here: Sting frequency increases with the order of trials, you need to prove here that the reactions were consistent over the 3 cycles to show there is no cumulative effect. Abbreviation for second is s not sec.

We have updated the figure captions to include the appropriate time interval when dealing with sting or pinch frequency and we have changed the abbreviation for second. We have addressed the issue of possible cumulative effects in our response to point 2 above.

In terms of p-values, we have partly addressed this issue in point 3 above. We believe that p-values and significance testing does have value when interpreted properly. We agree that in this case the CIs are more conservative. However, it is important to strike a balance between reporting relationships that might not be real (type 1 error) and being too conservative and failing to report legitimate relationships (type 2 error). In this case we hope to strike that balance by including both Cis and the p-values from the post-hoc tests in these figures.

Figure 3. Difficult to assess the results without knowing the order of experiments.

We have tried to clarify the order of treatments in newly added results section 5.4

Figure 4. Did you use here marginal means and not arithmetic means? Why? The number of stings for example can easily be calculated as a mean for the 18 individuals. It does not look that there is a difference between sexes as all CIs overlap. It is also not possible to know which sex has which symbol.

We have updated this figure to include a legend showing the symbols for males and females. And we address the issue of CIs and p-values above.

We chose to show marginal means over the arithmetic means because the marginal means are based on the mixed models and adjust for the covariates included in the models (e.g. scorpion length, observation time, etc.; see section 5.8.2). As such we feel that these marginal means better show the underlying relationships over unadjusted arithmetic means.

Figure S1. I fail to see the point of this figure, what do you want to show by these plots?

We hope that our addition to the beginning of the results section (lines 126-130) adds clarity as to why this plot is included.

Round 2

Reviewer 3 Report

The revision certainly improved the manuscript, but in my opinion the analyses are not sufficiently transparent to assess. My main concern is unclarity about sample size used in the analyses. Your sample size should be in most cases 9 (if you distinguish the sexes) or 18, so how can in line 143 the sample size be 43? This is not possible without autocorrelation (pseudoreplication) as you have maximal 18 individuals (no test can have a higher n). If you test an individual three times, n is still 1, you need to e.g. average the values before further analysis. This is probably also the reason why most of the time you get significant test results for largely overlapping confidence intervals. This is not possible, around P = 0.05, you usually get slightly overlapping CIs, and at low P values, CIs never overlap. I therefore assume that inflated sample sizes in the tests biased the results towards higher significance. You also need to increase clarity in the analysis methods. For example, in line 562 you write you used four models, but it is not clear which models. You need to provide precise information which parameter you used as independent variables and in which way (e.g. fixed factor, random factor, covariate). In line 572 you write about 3 additional models, but which?

I think there is also still overuse of significance tests, e.g. in line 135, you provide averages of length for females and males (with CI would be better than SE), these differences are so evident that the test results are redundant. Besides, do not repeat results in text and figure, so either provide these results in the text (in the case for scorpion length more appropriate) or the figure (not important for the purpose of your study and not sufficiently labelled, for example the grey line around the points is not defined). If you measured length based on video recordings (line 539-541), then the measures are very approximative (did you correct for spatial distortion?), so definitely not worth providing results more precise than to the next mm.

You wrote in the rebuttal letter that you did not use the results of the first cycle in the analyses, but I could not find any mention of this in the methods. Anyways, I do not think this is a good way to analyse, especially as you wrote that in the beginning there was a cumulative effect and then some kind of habituation. Therefore, I think you should present each cycle independently (e.g. a vertical arrangement of charts for each cycle), so that the reader can actually see how the effect is in time. This will not have any effect on sample size as the sample size is the number of individuals, not the number of experiments.

I think there is now a bit too much repetition of methods in the results. You do not need to repeat the methods in the results, you just need to write the results in a way the reader can understand without reading the methods.

A few more detailed comments:

You write in line 493 that you used 4 testing arenas, so you need to add arena ID in the models to show there was no effect.

Line 152 and elsewhere. I doubt that you can reliably measure time to a hundredth of a second, one decimal is completely enough. Abbreviation for second is s not sec (still on some places in the manuscript).

Line 594. Data is plural, therefore these data, not this data

Author Response

Comment 1: My main concern is unclarity about sample size used in the analyses. Your sample size should be in most cases 9 (if you distinguish the sexes) or 18, so how can in line 143 the sample size be 43?

We have removed the “N =” and reworded line 154 in order to help clarify what we are referring to. In the context, we are simply describing the number of trials (43 out of 52) where we observed the scorpion for the full 90 seconds. We did not mean to suggest that 43 was a sample size for any analysis.

Comment 2: This is not possible without autocorrelation (pseudoreplication) as you have maximal 18 individuals (no test can have a higher n). If you test an individual three times, n is still 1, you need to e.g. average the values before further analysis. This is probably also the reason why most of the time you get significant test results for largely overlapping confidence intervals. This is not possible, around P = 0.05, you usually get slightly overlapping CIs, and at low P values, CIs never overlap. I therefore assume that inflated sample sizes in the tests biased the results towards higher significance.

We agree that N = 17 or 18 for this study, however, we disagree with the charge that our analysis suffers from autocorrelation or pseudoreplication. We utilized linear and generalized linear mixed models which are designed to account for data that lacks independence and are commonly used to analyze repeated measures designs (see West, B. T., Welch, K. B., & Galecki, A. T. (2014). Linear mixed models: a practical guide using statistical software. CRC Press.; Harrison, X. A., Donaldson, L., Correa-Cano, M. E., Evans, J., Fisher, D. N., Goodwin, C. E., ... & Inger, R. (2018). A brief introduction to mixed effects modelling and multi-model inference in ecology. PeerJ6, e4794.)

Also, while we agree that some of the pair-wise comparisons we flag as statistically significant do show overlapping 95% confidence intervals in our figures, this does not mean that these tests are biased. This is because the pair-wise test is based on a single distribution representing the difference between group means rather than by comparing the distributions of the groups separately (see https://towardsdatascience.com/why-overlapping-confidence-intervals-mean-nothing-about-statistical-significance-48360559900a, and https://statisticsbyjim.com/hypothesis-testing/confidence-intervals-compare-means/ for more explanation). This means that 95% CI can overlap more than just a little bit and the related hypothesis testing can still generate a p-value that is less than 0.05. The other thing to keep in mind here is that many of the models we use are based on Poisson distributions rather that normal distributions. This may affect the amount of overlap in the 95% CIs as well.

Comment 3: You also need to increase clarity in the analysis methods. For example, in line 562 you write you used four models, but it is not clear which models. You need to provide precise information which parameter you used as independent variables and in which way (e.g. fixed factor, random factor, covariate). In line 572 you write about 3 additional models, but which?

We have updated the wording (lines 601-604, 606-607, and 616-617) to include more clarity about which variables are dependent or independent. Information about which variables are fixed factors or covariate controls was already present. Since the random factors used in all the models was the same, we include a single statement about them in lines 610-611.

Comment 4: I think there is also still overuse of significance tests, e.g. in line 135, you provide averages of length for females and males (with CI would be better than SE), these differences are so evident that the test results are redundant. Besides, do not repeat results in text and figure, so either provide these results in the text (in the case for scorpion length more appropriate) or the figure (not important for the purpose of your study and not sufficiently labelled, for example the grey line around the points is not defined).

In order to reduce redundancy, we have removed the means and SEs of the lengths of male and female scorpions (line 146-147). Since readers may be less familiar with violin plots (grey lines) we have updated figure 1 to include boxplots instead. We agree that the differences in length between males and females is obvious, however, we retain the results from the Student’s t-test to be consistent with our use of hypothesis testing in the rest of the manuscript.

Comment 5: If you measured length based on video recordings (line 539-541), then the measures are very approximative (did you correct for spatial distortion?), so definitely not worth providing results more precise than to the next mm.

You are correct in that these lengths are approximate. We did not correct for special distortion and have added wording to the methods to specify that fact (line 584). As described above, we no longer report mean lengths for males and females so the significant digits of these measurements are no longer relevant.

Comment 6: You wrote in the rebuttal letter that you did not use the results of the first cycle in the analyses, but I could not find any mention of this in the methods. Anyways, I do not think this is a good way to analyse, especially as you wrote that in the beginning there was a cumulative effect and then some kind of habituation. Therefore, I think you should present each cycle independently (e.g. a vertical arrangement of charts for each cycle), so that the reader can actually see how the effect is in time. This will not have any effect on sample size as the sample size is the number of individuals, not the number of experiments.

The description of omitting cycle 1 from analysis in some of the models is included in the methods (see lines 612-614).

The main point of our research was primarily to look at the effects of prod location, awareness of the local environment, and sex on defensive behavior. Many of the other variables we include in the analysis, including cycle, are secondary. We have done our best to control for the effects of these secondary variables on the relationships we were primarily interested in. Controlling for cycle was somewhat more challenging. Originally, our intention was to control for the possible effects of cycle by including it in some of our models. However, this ended up not being workable because we couldn’t get the models to converge making then largely uninterpretable (see lines 564-565). Our approach was to omit cycle one from these analyses since this is the first time the scorpions were exposed to the stimulus thereby controlling for this possible effect, and then summing the number of stings, pinches, venom use, and flee duration across cycles two and three before including them in our statistical models. We feel this should at least help mitigate possible confounding due to cycle in these models because it essentially treats cycles two and three as a single unit in the analysis. This should remove any effect of cycle in these models. We feel that the suggestion to analyze each cycle independently does have some merit. However, we feel it also has a major drawback. In order for us to continue to work within the context of LMMs and GLMMs, implementing this would require us to greatly increase our number of models we would need. Instead of four models, one model for each dependent variable (sting frequency, pinch frequency, venom use, and flee duration) we would need 12 models, one for each of the four dependent variables across three cycles. This would increase the family-wise error rate and increase our use of hypothesis testing. While we have opted not to correct for family-wise error rate for our post-hoc tests, we feel that this increase in the number of models may push things a bit too far.

Comment 7: I think there is now a bit too much repetition of methods in the results. You do not need to repeat the methods in the results, you just need to write the results in a way the reader can understand without reading the methods.

We have changed some wording to simplify and reduce redundancy in the description of our methods within the results section, see lines132-133 and 152. We feel that what remains is the minimum necessary to understand the results without reading the methods first.

A few more detailed comments:

Comment 8: You write in line 493 that you used 4 testing arenas, so you need to add arena ID in the models to show there was no effect.

            It is not possible to include arena ID in the model, and we argue below that it is also not necessary. As to the possibility of including arena ID, we did not keep track of which arenas were used for each trial, thus we do not have the information necessary to include it in our statistical models. We also have no way of deducing this information from the recordings as arenas were identical.

As to the need to include arena ID in the model, we took several steps (cleaned arenas with 70% ethanol between trials, discarded and replaced substrate between trials, and used new refuges for each trial) to control for the repeated use of arenas, see lines 531-534. Because we attempted to control for the repeated use of arenas experimentally, we do not believe it is necessary to also control for arenas statistically.

Comment 6: Line 152 and elsewhere. I doubt that you can reliably measure time to a hundredth of a second, one decimal is completely enough. Abbreviation for second is s not sec (still on some places in the manuscript).

            All recordings made of defensive behaviors were shot at 30 fps, and all behaviors were measured frame by frame. Significant digits are defined as all digits known with certainty and the first uncertain or estimated digit. Thus, in our case we can be certain of tenths place but not of the hundredths place. Therefore, we have changed all reporting of behaviors in seconds to the hundredth place. See lines 157.

Comment 7: Line 594. Data is plural, therefore these data, not this data

            Thank you for finding this. We have searched for our use of “data” and made sure that it is used as a plural throughout the manuscript. See change made on Line 570.

Round 3

Reviewer 3 Report

The manuscript has improved in the last revision. The authors decided not to follow my recommendation to present the results of individual cycles, but this is the decision of the authors and as they justify their choice in the manuscript, the readers can assess by themselves if this is a good way. However, figure 5 illustrates that you get significant results when both averages and confidence intervals show small differences. This does not give great confidence in the (plenty) other significance testing results, but I already pointed to this problem in my previous review. However, as the authors also think that the role of sex is not clear, it might be a better way not to use sex but directly size of individuals as continuous variable for the analyses. I also recommend another proofread for superfluous discussion (especially the multiple mention of “first time” at the end of discussion and the conclusion sounds pretentious and is unnecessary).

Author Response

Responses to Reviewer 3:

Comment 1: The manuscript has improved in the last revision. The authors decided not to follow my recommendation to present the results of individual cycles, but this is the decision of the authors and as they justify their choice in the manuscript, the readers can assess by themselves if this is a good way.

We want to thank reviewer 3 for the time they have spent giving feedback on our paper. Of all our reviewers, reviewer 3 has challenged us the most and caused us to carefully reconsider our results and methodologies. We recognize that this is the how good science works and we appreciate reviewer 3’s commitment to this process.

Comment 2: However, figure 5 illustrates that you get significant results when both averages and confidence intervals show small differences. This does not give great confidence in the (plenty) other significance testing results, but I already pointed to this problem in my previous review.

In many ways we share this concern. Showing statistically significant results while also showing widely overlapping confidence intervals should make the reader aware of the tentative nature of some of our results. We have explained our rational for the statistical decisions we have made in or previous responses to reviewer comments - especially as it relates to the balance that must be navigated between being too conservative in our conclusions (increasing the possibility of type 2 error) and being too liberal (increasing type I error). We leave it to the reader to decide whether we have erred one way or the other.

Comment 3: However, as the authors also think that the role of sex is not clear, it might be a better way not to use sex but directly size of individuals as continuous variable for the analyses.

We assume the reviewer is referring to our discussion of the effect of sex on defensive behavior and, in particular, lines 382-384 where we state that the effect of sex is still unclear. Our major point in that paragraph is to say that the effect of sex is unclear because of the potentially confounding nature of scorpion size, especially in species where there is significant size dimorphism. We have reworded these lines to make that clearer.

Since the major obstacle to clarity on the effect of sex on defensive behavior is disentangling it from the effect of size, we decided to include both size and sex in our statistical models. This means that our results related to sex, particularly with respect to venom use, should be interpreted as independent of size. Including results from models that include both size and sex may be helpful as others consider this question even though we feel our results are not conclusive, especially given the potential for bias we describe in lines 336-339.

The strong relationship between sex and size is something that we looked at when we developed our statistical models. Including strongly correlated independent variables within the same statistical model can lead to issues with multicollinearity, which can increase type 2 error and makes the estimates of coefficients unreliable. We examined this possibility using generalized variance inflation factors and decided that there wasn’t sufficient cause to omit sex or size from our models.

Comment 4: I also recommend another proofread for superfluous discussion (especially the multiple mention of “first time” at the end of discussion and the conclusion sounds pretentious and is unnecessary).

We have edited the discussion and conclusion sections to reduce redundancy. We have made changes to lines 398-399, 561-564, 581, 591, 596, 612-613, 630-631, and 634-635. We hope this makes the discussion sound less boastful and, instead, emphasizes the tentative nature of some of our results. We have specifically removed the repetition of the phrase “first time” as suggested. It was not our intention to sound pretentious, but to caution readers that our results concerning how refuges may have impacted venom use should be considered tentative and need further study to verify. We recognize that “early” results in areas that have little to no research should be viewed more skeptically and we intended our original language to emphasize that fact.